

1  Modeling Regional Air Quality and Climate: Improving Organic Aerosol and Aerosol Activation
2                          Processes in WRF/Chem version 3.7.1

Khairunnisa Yahya[1], Timothy Glotfelty[1], Kai Wang[1], Yang Zhang[1]*, and Athanasios Nenes[2,3,4,5]

[1]Department of Marine, Earth, and Atmospheric Sciences, North Carolina State University,
Raleigh, North Carolina, U.S.A.
[2]School of Earth and Atmospheric Sciences, Georgia Institute of Technology, Atlanta, GA,
U.S.A
[3]School of Chemical and Biomolecular Engineering, Georgia Institute of Technology, Atlanta,
GA, U.S.A.
[4]Institute of Environmental Research & Sustainable Development, National Observatory of
Athens, Greece
[5]Institute for Chemical Engineering Science, Foundation for Research and Technology-Hellas,
Patra, Greece
Email: *yang_zhang@ncsu.edu

**ABSTRACT**

Air quality and climate influence each other through the uncertain processes of aerosol formation
and cloud droplet activation. In this study, both processes are improved in the Weather, Research
and Forecasting model with Chemistry (WRF/Chem) version 3.7.1. The existing Volatility Basis
Set (VBS) treatments for organic aerosol (OA) formation in WRF/Chem is improved by
considering the secondary OA (SOA) formation from semi-volatile primary organic aerosol
(POA), a semi-empirical formulation for the enthalpy of vaporization of SOA, as well as
functionalization and fragmentation reactions for multiple generations of products from the
oxidation of VOCs. Two-month long simulations (May to June 2010) are conducted over
continental U.S. and results are evaluated against surface and aircraft observations during the
Nexus of Air Quality and Climate Change (CalNex) campaign. Among all the configurations
considered, the best performance is found for the simulation with the 2005 Carbon Bond
mechanism (CB05) and the VBS SOA module with semivolatile POA treatment, 25%



fragmentation, and the emissions of semi-volatile and intermediate volatile organic compounds
being 3 times of the original POA emissions. Among the three gas-phase mechanisms (CB05,
CB6, and SAPRC07) used, CB05 gives the best performance for surface ozone and $PM_{2.5}$
concentrations. Differences in SOA predictions are larger for the simulations with different VBS
treatments (e.g., non-volatile POA vs. semivolatile POA) as compared to the simulations with
different gas-phase mechanisms. Compared to the simulation with CB05 and the default SOA
module, the simulations with the VBS treatment improve cloud droplet number concentration
(CDNC) predictions (NMBs from -40.8% to a range of -34.6% to -27.7%), with large differences
between CB05/CB6 and SAPRC07 due to large differences in their OH and $HO_2$ predictions. An
advanced aerosol activation parameterization based on the FN05 series reduces the large negative
CDNC bias associated with the default ARG00 parameterization from -35.4% to a range of -0.8%
to 7.1%, it, however, increases the errors due to overpredictions of CDNC, mainly over
northeastern U.S. This work indicates a need to improve other aerosol-cloud-radiation processes
in the model such as the spatial distribution of aerosol optical depth and cloud condensation nuclei
in order to further improve CDNC predictions.
**1. Introduction**
The Intergovernmental Panel on Climate Change (IPCC) report on the AR5 scenario attributes
the aerosol radiative forcing (RF) to be the dominant source of uncertainty contributing to the
overall uncertainty in the net Industrial Era Radiative Forcing (RF) calculations (Myhre et al.,
2013). Despite the inclusion of more aerosol processes in the current generation of atmospheric
models, differences between atmospheric models and observations continue to persist. Aerosols
affect the climate through the direct effect by absorbing or scattering radiation, or the indirect
effect by acting as cloud condensation nuclei (CCN).  According to Hallquist et al. (2009), the



formation of inorganic particulates such as sulfate, nitrate, and ammonium are well understood,
however, there are large uncertainties in the formation of secondary organic aerosol (SOA). As a
result, current models do not have a comprehensive treatment of OA, which usually result in an
underprediction of OA concentrations (Hodzic et al., 2010; Jathar et al., 2011; Bergstrom et al.,
2012), due to missing key precursors and processes in OA formation (Ahmadov et al., 2012). Some
of the missing key precursors in most models include semi-volatile primary organic aerosol (POA),
long-chain *n*-alkanes, polycyclic aromatic hydrocarbons (PAHs), and large olefins that have lower
volatilities compared to traditional SOA precursors (Chan et al., 2009). The organic carbon (OC)
component of the radiative forcing in the IPCC AR5 report also does not include SOA with the
reason that the formation is dependent on a number of factors that are not currently sufficiently
quantified (Myhre et al., 2013). However, SOA can form a significant percentage of total OA (up
to 95% in rural areas) (Zhang et al., 2007). Another large source of uncertainty is the quantification
of clouds as well as aerosol-cloud interactions due to incomplete or inaccurate representations of
these processes in climate models (Boucher et al., 2013). A major process in cloud formation from
aerosol is aerosol activation, which involves the condensational growth of aerosols in a cooling air
parcel until maximum supersaturation, and some of the wet particles reach a critical radius where
they are then able to grow spontaneously into cloud droplets (Ghan et al., 2011). Various
approaches have been developed to reduce the uncertainties associated with OA and aerosol
activation treatments in climate models. Those treatments are reviewed in the following section.
**1.1. VBS Treatments and Sensitivity to Different Gas-Phase Chemical Mechanisms in**
74         **Regional and Global Models**
Unlike inorganic aerosols such as sulfate, the physical and chemical properties of OA

dynamically evolve with age (Jimenez et al., 2009). The traditional approach to modeling SOA is



to assume that each VOC precursor forms several surrogate compounds (Odum et al., 1996).
However, the traditional method has several shortcomings, for example, two products are needed
for each VOC precursor causing this method to be computationally-expensive if many VOC
precursors are treated in the model (Murphy and Pandis, 2009). The assumption that the products
are unreactive also does not reflect the dynamic nature of the first generation products from the
oxidation of VOCs that can undergo successive oxidation steps to further produce lower volatility
products (Jimenez et al., 2009). The volatility basis set (VBS) is a framework developed by
Donahue et al. (2006), which is able to simulate gas-phase partitioning and multiple generations
of gas-phase oxidation of organic vapors. This approach addresses the shortcomings of the
traditional SOA modeling approach as it can cover the complete volatility range of OA compounds
(Murphy and Pandis, 2009).
Table 1 summarizes some of the VBS treatments from current regional and global models. The
VBS treatment has been implemented into a number of regional models such as the Weather,
Research and Forecasting model with Chemistry (WRF/Chem)  (Shrivastava et al., 2011;
Ahmadov et al., 2012),the Particulate Matter Comprehensive Air Quality Model with extensions
(PMCAMx) (Lane et al., 2008; Donahue et al., 2009; Murphy et al., 2009), and CHIMERE (Hodzic
et al., 2010). It has also been implemented in global models such as GISS II' GCM (Farina et al.,
2010; Jathar et al., 2011) and the Community Earth System Model (CESM) (Shrivastava et al.,
2015). Different studies define the classifications of the organic species slightly differently.
Donahue et al. (2009) defined primary organic vapors with effective saturation concentrations (C*)
of $10^{-2}$ - $10^{-1}$, $10^0$ - $10^2$, and $10^3 - 10^6$ µg m$^{-3}$at 298 K to be low volatility organic compounds
(LVOCs),semi-volatile organic compounds (SVOCs), and intermediate volatility organic
compounds (IVOCs), respectively.  Shrivastava et al., (2011) and Jathar et al. (2011) defined



primary organic vapors with C* values of $10^{-2}$ - $10^3$ and $10^4 – 10^6$ µg m$^{-3}$ to be SVOCs and IVOCs,
respectively. All those studies defined VOCs to be gas-phase organic species with C* larger than
$10^6$ µg m$^{-3}$ at 298 K.
The traditional emission inventories used in the chemical transport models consist of VOCs
but not SVOCs or IVOCs as both SVOCs and IVOCs are difficult to measure. This is most likely
because SVOCs and IVOCs tend to evaporate at high temperatures from combustion sources
(Donahue et al., 2009). As the traditional SOA approach usually underpredicts the SOA
concentration, the addition of the SVOC and IVOC emissions on top of the existing VOC
emissions in most emission inventories can improve model performance. To account for the
missing key precursors in OA formation, SVOC and IVOC emissions are usually estimated as a
factor of existing POA emissions in current emission inventories. For example, Shrivastava et al.
(2011) estimated the sum of all SVOC and IVOC precursors to be 7.5 times the mass of traditional
POA emissions inventory over Mexico City, but indicated that the scaling factor of 3 for SVOC
emissions based on the POA emissions is poorly constrained. Shrivastava et al. (2008) and Jathar
et al. (2011) assumed that SVOC emissions are represented by the traditional emission inventory
while IVOC emissions are 1.5 times the traditional emission inventory. Pye and Seinfeld (2010)
assumed that SVOC emissions are a subset of traditional POA emission inventories, and their POA
emissions were scaled up by 27% on a global scale. IVOC emissions are assumed to be spatially-
distributed similar to naphthalene and are predicted to be roughly a factor of half of global POA
emissions. Tsimpidi et al. (2014) assumed that the IVOC emissions are 1.5 times the traditional
POA emission inventory and are assigned to the 4$^{th}$ volatility bin with C* = $10^5$ µg m$^{-3}$. For
comparison, some studies such as Ahmadov et al. (2012) and Bergstrom et al. (2012) used the
VBS approach for OA modeling but did not include additional SVOC emissions. There are also





differences in the volatility distribution used in literature. Shrivastava et al. (2008) and Jathar et
al. (2011) found that moving half the mass of SVOC from all bins to the lowest bin from the
traditional "diesel exhaust" volatility distribution of Robinson et al. (2007) produced the lowest
errors in simulated OA on an annual average basis.

The number of bins used can also result in differences in simulated SOA concentrations.

Shrivastava et al. (2011) showed that the 2-species VBS performed better than the 9-species VBS
in modeling oxygenated organic aerosol (OOA) and gave the closest agreement to the OOA
calculated by the Positive Matrix Factorization (PMF) method. This indicates that SOA may be
less volatile as compared to the volatility distribution in the 9-species VBS which allows for
evaporation of SOA with dilution (Shrivastava et al., 2011).

The amount of oxygen added for each oxidation step may contain uncertainties. This factor

can influence the O:C ratio used for the model evaluation. O:C predictions from models need to
be improved by including fragmentation reactions (which could lead to an increase in O:C ratios)
and improving emission estimates (Shrivastava et al., 2011). Different rate constants can also result
in different predictions of SOA concentrations. For example, Farina et al. (2010) showed that the
use of k value of $1\times10^{-12}$ cm$^3$ molecule$^{-1}$ s$^{-1}$ compared to the default k value of $10\times10^{-12}$ cm$^3$
molecule$^{-1}$ s$^{-1}$ resulted in a reduced aged SOA formation by 71%. Hodzic et al. (2010) also showed
a case study based on Grieshop et al. (2009) in which each oxidation step reduced the volatility of
the S/IVOC vapors by two orders of magnitude and each successive oxidation step produced a
40% increase in mass due to the addition of oxygen. This case is inconclusive in urban areas - a
larger bias along with a higher correlation coefficient compared to the more common case where
each oxidation step reduced the volatility by one order of magnitude with a 7.5% increase in mass.



However, the model performed worse (with larger bias and lower correlation coefficient) in
suburban areas.
The aging process improves model performance in general in the United States (U.S.) but
deteriorates the performance in several parts of Europe. Accounting for the aging process of OA
will increase the OA concentrations and improve model results in the U.S. where OA is usually
underpredicted, but increase the model bias for OA in several parts of Europe where OA
concentrations are overpredicted (Farina et al., 2010; Bergstrom et al., 2012).
Shrivastava et al. (2013) studied the effects of the fragmentation and functionalization in VBS.
Functionalization increases the mass of OA for each successive oxidation step, while
fragmentation reduces the mass for each oxidation step. One such a case includes simulating first-
order effects of the fragmentation and functionalization processes in VBS by assuming
functionalization of 100% of organic vapors for the first two generations of oxidation and both
fragmentation and functionalization for the third and higher generations of oxidation. The
fragmentation reduces the SOA concentrations drastically. For example, Shrivastava et al. (2013)
showed that peak SOA concentrations can be reduced by factors of 2 to 4 for a 1-hour example on
10 March 2006 at 21 UTC over Mexico City Plateau.
The VBS framework for OA modeling in the latest version of WRF/Chem, v3.7.1, is coupled
with several gas-phase mechanisms including the 2005 Carbon Bond Mechanism (CB05)
(Yarwood et al., 2005), the Model for Ozone and Related chemical Tracers version 4 (MOZART-
4) (Emmons et al., 2010), the Regional Atmospheric Chemistry Model (RACM) (Stockwell et al.,
1997), and the 1999 version of the Statewide Air Pollution Research Centre (SAPRC99)
mechanism (Carter, 2000). Different gas-phase mechanisms have different lumpings/groupings
for VOCs, which will affect OA formation. For example, VOCs are lumped according to their



carbon bonds (e.g., single or double bond) in CB05 (Yarwood et al., 2005) while VOCs in
SAPRC99 (Carter, 2000) are lumped according to their OH reactivities. A number of studies
examined the differences in predicting $O_3$ concentrations due to different gas-phase mechanisms
(e.g., Luecken et al., 2008; Li et al., 2012; Shearer et al., 2012; Zhang et al., 2012), but fewer
studies reported the impact of different gas-phase mechanisms on modeling SOA and $PM_{2.5}$
concentrations (Kim et al., 2011; Zhang et al., 2012). SAPRC99 has more detailed organic
chemistry compared to CB05. SAPRC99 has been updated to SAPRC07 (and recently, to
SAPRC11) based on newly available information regarding the reactions and influence of
individual VOCs on $O_3$, as well as evaluations against chamber experiments (Carter, 2010). In
addition, SAPRC07 has reformulated reactions of peroxy radicals so that the effects of changes in
nitrogen oxides ($NO_x$) on organic product formation is more accurately represented. SAPRC07
has the most extensive set of VOC species and reactions, as compared to CB05 and the Carbon
Bond version 6 (CB6). Shearer et al. (2012) reported that a condensed version of SAPRC07
predicted lower $O_3$ and OH concentrations in central California compared to SAPRC99 due to a
decreased reaction rate coefficient in the reaction of OH and $NO_2$ to form $HNO_3$. Li et al. (2012)
also showed that predicted $O_3$ concentrations from SAPRC07 were lower than those of SAPRC99
by up to 20% over Texas. The same study also reported that SAPRC07 gave lower OH
concentrations due to differences in the reaction rate constants in the reactions of $O^1D$ and $H_2O$
between SAPRC07 and SAPRC99. Luecken et al. (2008) reported that SAPRC99 gave higher $O_3$
concentrations compared to CB05 on average; however, the differences vary with locations,
VOC/$NO_x$ ratios, and the concentrations of precursor pollutants. This is consistent with the results
from Zhang et al. (2012), which predicted that SAPRC99 using WRF/Chem with the Model of
Aerosol, Dynamics, Reaction, Ionization and Dissolution (WRF/Chem-MADRID) produced the





highest O$_3$ mixing ratios in July at the Southeastern Aerosol Research and Characterization
(SEARCH) sites. The CB6 (Yarwood et al., 2010) is an updated version of CB05 with improved
kinetic and photolysis data, additional explicit species for long-lived and abundant organic
compounds including propane, acetone, benzene and acetylene, as well as revised isoprene and
aromatics chemistry from CB05. Yarwood et al. (2010) showed that CB6 produces higher daily
maximum 8-hr O$_3$ as compared to CB05 over Los Angeles for one episode day in August with the
highest observed O$_3$ mixing ratios. CB6 was also shown to produce substantially higher OH
concentrations (25% to 50% higher at mid-day over large areas) over eastern U.S. compared to
CB05 over a few days in June, 2006. A summary of the main characteristics of CB05, CB6, and
SAPRC07 gas-phase mechanisms are listed in Table 2.
**1.2. Description of Aerosol Activation Parameterizations**

Ghan et al. (2011) provided a comprehensive review on various aerosol activation treatments

in current climate models. Two main types of parameterizations are commonly used: the Abdul-
Razzak and Ghan (2000) (AR-G00) and the Fountoukis and Nenes (2005) (FN05) and associated
updates described in Barahona et al. (2010) and Morales Betancourt and Nenes (2014). AR-G00
uses multiple lognormal or sectional distributions to approximate the aerosol size distribution. It
uses the Kohler theory to relate the aerosol size distribution and composition to the number of
aerosols activated as a function of maximum supersaturation (S$_{max}$). The complex function
involving S$_{max}$ is parameterized based on the standard deviation σ from a large number of
numerical solutions using a cloud parcel model. The number and mass activated are particles with
critical supersaturation less than S$_{max}$. It also accounts for particle growth before and after the
particles are activated. However, the ARG treatment does not explicitly represent kinetic
limitations which tend to affect smaller or larger particles (with diameters far from their critical



size). Very small particles tend to lose water when supersaturation declines as they never exceed
the critical supersaturation for that particle size, and very large particles may not have achieved
the critical size before $S_{max}$ is reached (Ghan et al., 2011).  Kinetic limitations refer to the (i) inertial
mechanism – where particles with large dry diameters grow to be as large as activated particles
but have not been activated themselves, these particles should be considered together with
activated particles; (ii) evaporation mechanism – where particles with high critical supersaturation
evaporate before reaching their critical diameters; and the (iii) deactivation mechanism – where
initially activated particles that are deactivated to interstitial aerosols when the parcel
supersaturation falls below the equilibrium supersaturation (Nenes et al., 2001). Neglecting kinetic
limitations performs well for all conditions except in highly-polluted areas (Ghan et al., 2011). In
urban and highly-polluted cases, many particles fail to be activated due to strong evaporation and
deactivation processes (Nenes et al., 2001). Explicitly accounting for kinetic limitations reduces
CDNC at low updraft velocity (Ghan et al., 2011).
The Fountoukis and Nenes (2005) (FN05) scheme improved the ARG00 scheme by solving
$S_{max}$ analytically (with the exception of kinetically-limited particles) using a so-called "population
splitting" method. In addition, FN05 took into account the kinetic limitations, as well as the
influence of gas kinetics on water vapor diffusivity (Ghan et al., 2011). The other improved
treatments built on top of the FN05 scheme include the entrainment of ambient air, which could
reduce the supersaturation of the updraft (Barahona and Nenes, 2007) (BN07) (therefore reducing
CDNC); the adsorption of water vapor onto insoluble particles by Kumar et al. (2009) (KU09)
based on a modified Frenkel-Halsey-Hill (FHH) adsorption theorem (which will increase CDNC);
the growth of giant cloud condensation nuclei (CCN) (Barahona et al., 2010) (BA10) by
introducing an additional condensation rate term to account for condensation of giant CCN (which



will reduce CDNC); as well as the modification of the original population splitting concept in
FN05 and BA10 by Morales Betancourt and Nenes (2014) (MN14) by better accounting for the
size of inertially limited CCN, and removing a discontinuity in the calculation of the surface area
of cloud droplets.
The parameterization of Abdul Razzak and Ghan (2000) (ARG00) is used as the default
aerosol activation module in WRF/Chem. It is not linked to the microphysics module or cumulus
parameterization in WRF or WRF/Chem. However, for WRF/Chem, the cloud droplet number
concentration (CDNC) generated in ARG00 is passed to the microphysics scheme, i.e., the
Morrison two-moment microphysics scheme selected in this work.
**1.3 Motivations and Objectives**
The online-coupled meteorology and chemistry model, WRF/Chem, has recently been
applied for air quality and climate modeling on a decadal scale (Yahya et al., 2016a, b).
WRF/Chem can also simulate aerosol direct and indirect feedbacks, which are important
considerations for climate modeling. However, as mentioned previously, the representations of
OA and aerosol-cloud interactions in current regional and global climate models are subject to
large uncertainties. In particular, while the VBS framework in WRF/Chem significantly improves
SOA performance (Wang et al., 2015), it lacks the semi-volatile POA treatment, as well as
fragmentation processes (Shrivastava et al., 2013). The first objective of this study is to reduce
uncertainties associated with OA predictions by improving the existing VBS module in
WRF/Chem and identifying the best gas-phase chemical mechanism to drive the VBS module for
the most accurate OA predictions. The impact of the improved OA predictions on CDNC in
WRF/Chem will be quantified. The second objective is to incorporate an improved aerosol



activation parameterization based on the FN05 series into WRF/Chem to study its impacts on
CDNC predictions.
**2.   Model Configuration, Evaluation Protocol, and Observational Datasets**
**2.1.Model Setup and Inputs**
The model used in this study is a modified version of WRF/Chem v3.7.1 as described by Wang

et al. (2015). The 2005 Carbon Bond gas-phase mechanism (CB05) of Yarwood et al. (2005) with
additional chlorine chemistry is coupled with the Modal for Aerosol Dynamics in Europe −
Secondary Organic Aerosol Model (MADE/SORGAM) (Ackermann et al., 1998; Schell et al.,
2001) and the Volatility Basis Set (MADE/VBS) (Ahmadov et al., 2012). The CB05-VBS option
has also been coupled to existing model treatments including the aerosol direct effect, the aerosol
semi-direct effect on photolysis rates of major gases, and the aerosol indirect effect on CDNC and
resulting impacts on shortwave radiation. The physics options used in WRF/Chem include the
rapid and accurate radiative transfer model for GCM (RRTMG) for both shortwave and longwave
radiation, the Yonsei University (YSU) planetary boundary layer (PBL) scheme (Hong et al., 2006;
Hong, 2010), the Morrison et al. (2009) double moment microphysics scheme, as well as the Multi-
scale Kain-Fritsch (MSKF) cumulus parameterization scheme (Zheng et al., 2016). Aqueous-
phase chemistry module (AQCHEM) for both resolved and convective clouds is based on a similar
AQCHEM module in CMAQv4.7 of Sarwar et al. (2011). The anthropogenic emissions used are
from the 2010 emissions based on the 2008 U.S. Environmental Protection Agency (U.S. EPA)
National Emissions Inventory (NEI) from the Air Quality Model Evaluation International Initiative
(AQMEII) project (Pouliot et al., 2015). Dust emissions are based on the Atmospheric and
Environmental Research Inc. and Air Force Weather Agency (AER/AFWA) scheme (Jones and
Creighton, 2011). Emissions from sea salt are generated based on the scheme of Gong et al. (1997).





Biogenic emissions are simulated online by the Model of Emissions of Gases and Aerosols from
Nature v2.1 (MEGAN2.1) (Guenther et al., 2006).

The chemical initial and boundary conditions (ICONs/BCONs) come from the modified

CESM/CAM version 5.3 with updates by Gantt et al. (2014), He and Zhang (2014), and Glotfelty
et al. (2016). The meteorological ICONs/BCONs are from the National Center for Environmental
Protection Final Reanalyses (NCEP FNL) dataset, which is available every 6 hours. The chemical
fields are also allowed to run continuously while the meteorology is reinitialized every 5 days. The
simulations are performed at a horizontal resolution of 36-km with $148 \times 112$ horizontal grid cells
over the CONUS domain and parts of Canada and Mexico, and a vertical resolution of 34 layers
from the surface to 100-hPa.

A number of sensitivity simulations are designed to identify the model configuration with

results that are in the closest agreement to observations as well as the realistic model treatments of
OA that are the closest to atmospheric processes. The baseline and sensitivity simulations are
conducted from May to June 2010, during which the Nexus of Air Quality and Climate Change
(CalNex) campaign was held in Bakersfield and Pasadena, California. The first 10 days from May
$1^{st}$ to May $10^{th}$ are considered to be the spin-up period.
**2.2.    Model Evaluation Protocol and Available Measurements**
Statistical measures including the Mean Bias (MB), Correlation Coefficient (Corr),

Normalized Mean Bias (NMB) and Normalized Mean Error (NME) (Yu et al., 2006) are used to
evaluate the simulations against observational data. Observational data are available for organic
carbon (OC) and total carbon (TC) from the Speciated Trends Network (STN) and the Interagency
Monitoring for Protected Visual Environments (IMPROVE). While both OC and TC from
IMPROVE are used for model evaluation, only TC data from STN are used as STN uses the



thermo-optical transmittance protocol for OC that is different from the one used by IMPROVE
(Zhang et al., 2012). In addition, the measurements for STN OC are not blank corrected for carbon
on the background filter (Wang et al., 2012). The ratios OA/OC ratios vary across locations in the
continental U.S. (CONUS) depending on whether the OA is dominated by secondary formation
(Aitken et al., 2008) or it contains more aliphatic hydrocarbons (Turpin and Lim, 2001). In this
study, two ratios, 1.4 and 2.1, are used to convert simulated OA to OC based on a number of
literature (Turpin and Lim, 2001; Aitken et al., 2008; Xu et al., 2015). As the simulations are based
on CONUS with varying OA properties (less or more oxidized OA), the use of two OA/OC ratios
can represent the different types of OA present for all locations in the U.S. Spatial plots, time series
plots at specific sites, as well as overlay plots are used to evaluate model performance. The
IMPROVE sites chosen for the time series plots include the visibility-protected areas in Brigantine
National Wildlife Refuge (NWR), NJ, Death Valley National Park (NP), CA, Swanqwarter
National Wildlife Refuge (NWR), NC, and the Tallgrass Prairie National Preserve, KS. The
Brigantine NWR is a tidal wetland and has a shallow bay, the Death Valley NP is a desert, and the
Swanqwarter NWR is a coastal brackish marsh. The time series plots are made at four STN sites
including two urban sites: in Washington, DC and Boise, ID, one industrial site in Tampa, FL, and
one rural/agricultural site in Liberty, KS. SOA, hydroxyl radical (OH) and hydroperoxy radical
($HO_2$) data are also available for May to June 2010 as part of the California Research at the Nexus
of Air Quality and Climate Change (CalNex) campaign (Kleindienst et al., 2012; Lewandowski et
al., 2013) in Bakersfield, CA and Pasadena, CA, which are both urban locations. The Bakersfield
sampling site is located between the city center and areas of agricultural activity, while the
Pasadena site is located at the California Institute of Technology campus within the Los Angeles
metropolitan area to the southwest and mountains in the north (Baker et al., 2015).



POA/OA ratios are also used to evaluate the performance of the model. A number of studies
have reported observed POA/OA ratios which range from 15% to 40% over CONUS. For
example, over southeastern U.S., hydrocarbon-like OA (HOA) and cooking OA are found to
contribute to $21 - 38\%$ of total OA in urban sites (Xu et al., 2015). HOA and oxygenated OA
(OOA) are found to account for 34% and 66% of measured OA from Pittsburgh in September 2002
(Zhang et al., 2005). HOA and cooking OA are assumed to be synonymous to POA, and OOA is
assumed to be synonymous to SOA. Particulate matter sampled during August and September
2006 in Houston as part of the Texas Air Quality Study II Radical and Aerosol Measurement
Project showed that approximately 32% of OA comes from HOA (Cleveland et al., 2012). Results
from positive matrix factorization analysis from the Pasadena ground site during May and June
2010 showed that the primary components contribute 29% of the total OA mass (Hayes et al.,
2013). Based on Zhang et al. (2007), the percentages of HOA mass at urban sites in Riverside,
CA, from mid-July to mid-August 2005, in Houston, TX, from mid-August to mid-September
2000, and in New York City in July 2001 are 15%, 38%, and 30%, respectively. In addition, Zhang
et al. (2011) compiled a large number of field campaigns across the globe where the average
POA/OA ratios for urban, downwind and rural/remote areas are found to be 0.42, 0.18 and 0.10
respectively.
For the aerosol activation sensitivity and production simulations, additional variables that will
be analyzed in this study include maximum 1-hour and 8-hour $O_3$ against the Clean Air Status and
Trends Network (CASTNET) and Air Quality System (AQS), aerosol optical depth (AOD),
CDNC and cloud condensation nuclei (CCN) against MODIS.
**3. Model Development and Improvement**



A number of modifications have been made to the standard version of WRF/Chem model
v3.7.1. Those modifications and treatments are described below.
**3.1. OA Treatments**

The CB05-VBS treatment in the default WRF/Chem v3.7.1 assumes that POA is nonreactive
and nonvolatile. In this study, POA is assumed to be semivolatile, and can undergo gas-particle
partitioning, similar to anthropogenic SOA (ASOA) and biogenic SOA (BSOA) in VBS. While
the volatility of ASOA and BSOA is represented by 4 bins with C* from $10^0$ to $10^3$ μg m$^{-3}$. The
POA is distributed into 9 bins, with C* from $10^{-2}$ to $10^6$ μg m$^{-3}$, following the set-up of Shrivastava
et al. (2011). The POA is oxidized to form semi-volatile OA (SVOA), which can also undergo
gas-particle partitioning.  For the POA, bin-resolved enthalpies of vaporizations are used, ranging
from 64 kJ mol$^{-1}$ for the 9$^{th}$ bin to 112 kJ mol$^{-1}$ for the 1$^{st}$ bin according to Shrivastava et al. (2011).
The default enthalpy of vaporization ($\Delta H_{vap}$) for SOA in WRF/Chem is 30 kJ mol$^{-1}$ according to
Lane et al. (2008). A more accurate alternative is to use the $\Delta H_{vap}$ values calculated from the semi-
empirical correlation from Epstein et al. (2010):

$\Delta H_{vap} = -11 \log_{10} C^*_{300} + 129$                           (1)

The values of $\Delta H_{vap}$ Epstein et al. (2010) are used in a number of sensitivity simulations and final
production simulation.
Shrivastava et al. (2013, 2015) also implemented several cases of fragmentation and
functionalization (FF) processes into VBS. For this study, the FF set-up is similar to the method
employed by Shrivastava et al. (2013), with the exception that fragmentation percentages of 10%,
25%, and 50% are used in sensitivity simulations. Shrivastava et al. (2013) used fragmentation
percentages of 50% (intermediate fragmentation) and 85% (high fragmentation) in his simulations
over Mexico City. For example, for the 10% FF case, 10% of the mass in the VBS species is



functionalized and moved to the next lower volatility bin, 80% is fragmented and moved to the
highest volatility bin, and the remaining 10% is fragmented and becomes more volatile than the
highest volatility bin (i.e., it is lost from the current volatility bins). For the 50% FF case, 50% is
functionalized and moved to the next lower volatility bin, 40% is fragmented and moved to the
highest volatility bin, and 10% is lost.
Zhao et al. (2014) measured IVOCs in Pasadena, CA during CalNex and found that the
concentrations of primary IVOCs are similar to those of single-ring aromatics, and they produce
about 30% of newly formed SOA in the afternoon. With the semivolatile POA and FF cases in this
study, additional IVOC and SVOC emissions are added as three times of the traditional POA
emissions from NEI, to account for missing IVOC and SVOC species in the traditional POA
emission inventory. The fraction of IVOC/SVOC emissions assigned to each volatility bin is
summarized in Table 3.
The mass fraction of organics in each volatility bin determined in laboratory studies also differs
significantly according to the sources of organics. For example, May et al. (2013a, b, c) has
different volatility distributions of mass fractions of organics for gasoline vehicle exhaust, diesel
exhaust, and biomass burning. To take into account the different sources of organic compounds
into a single volatility distribution for the purpose of this work, a new volatility distribution is
calculated based on the mass fractions reported by Shrivastava et al. (2011), May et al. (2013a, c)
and the percentages of VOC emissions from various sources from the 2008 NEI. According to the
2008 NEI report (Rao et al., 2013), total VOC emissions from stationary, mobile and fire
(prescribed and wildfire) sources are ~7.6, ~5.6, and ~49.6 million tons, respectively. The
corresponding percentages for VOC emissions are ~12%, ~9%, and ~79% for stationary, mobile,
and fire sources, respectively. Based on the U.S. EPA (2013), the percentages of diesel emissions



from mobile sources are low compared to gasoline sources (~7% of total diesel and gasoline
sources); they are thus not included in this study.
An example calculation for the mass fraction of the lowest volatility bin for POA and
IVOC/SVOC emissions are as follows:
$\text{Log } C_{-2}*$ (at 298K) = 0.04×12% + 0.14×9%+0.79×79% = 0.1754                    (2)
where $C_{-2}*$ refers to the lowest volatility bin with a value of $10^{-2}$ μg m$^{-3}$, 12%, 9%, and 79% refer
to the percentages for VOC emissions from stationary, mobile, and fire sources, respectively from
NEI, 0.04 refers to the original mass fraction for stationary emissions based on anthropogenic
emissions from Shrivastava et al. (2011) for the lowest volatility bin with a value of $10^{-2}$ μg m$^{-3}$,
0.14 refers to the original mass fraction for gasoline emissions from May et al. (2013a) for the
lowest volatility bin with a value of $10^{-2}$ μg m$^{-3}$, 0.2 refers to the original mass fraction for biomass
burning emissions from May et al. (2013c) for the lowest volatility bin with a value of $10^{-2}$ μg m$^{-}$
$^{3}$, and 0.1754 refers to the newly-calculated mass fraction of POA and IVOC/SVOC emissions for
this study. The mass fractions used by Shrivastava et al. (2011), May et al. (2013a, c), and this
work can be found in Table 3.
**3.2.    Gas-Phase Chemical Mechanisms**
Three gas-phase mechanisms are used: CB05, CB6, and SAPRC07. The gas-phase
mechanisms for CB6 and SAPRC07 are coupled to the MADE/VBS in WRF/Chem v3.7.1 in this
work following the coupling of CB05 with MADE/VBS by Wang et al. (2014). The emissions for
all cases are based on the CB05 chemical species from the 2010 emissions based on the 2008 NEI.
For SAPRC07, slight modifications had to be made to account for the different VOC species or
groups. The mapping of emission species from CB05 to SAPRC07 is based on the grouping of




species from emitdb.xls from Henderson et al. (2014) as well as from
http://www.cert.ucr.edu/~carter/emitdb/old-emitdb.htm. CB05 emissions are used for the CB6
case, with the exception of the VOCs including propane, benzene, ethyne, acetone, and ketone that
are mapped based on fractions of existing CB05 VOCs according to Yarwood et al. (2010).
In VBS, the SOA precursors for CB6 are similar to those for CB05. The SOA precursors for
CB05 (and therefore CB6) are mapped from the default SAPRC99 precursors by Wang et al.
(2015). The SAPRC07 SOA precursors follow the existing mapping of SAPRC99-MOSAIC/VBS
in WRF/Chem. The chemical equations and rate parameters from ENVIRON (2013) and Carter
(2010) for CB6 and SAPRC07 gas-phase mechanisms, respectively, were included in the
chem/KPP/mechanisms directory in WRF/Chem. The SAPRC07 gas-phase mechanism
implemented in WRF/Chem in this case is the uncondensed and expanded version C, which
includes reactions for peroxy radical operators (Carter, 2010). Species in both CB6-MADE/VBS
and SAPRC07-MADE/VBS undergo dry deposition, aqueous chemistry, photolysis, and wet
scavenging that are similar to CB05-MADE/VBS.
**3.3. Aerosol Activation**
The FN05 series aerosol activation parameterizations (with the exclusion of MN14) have been
incorporated into 3-D regional air quality models and global climate and Earth system models such
as the WRF-Community Atmosphere Model version 5 (WRF-CAM5) (Zhang et al., 2015), and in
the global-through-urban WRF/Chem (GU-WRF/Chem) (Zhang et al., 2012) and CESM (Gantt et
al., 2014). In this study, the FN series parameterizations are incorporated into WRF/Chem
following the methods of Gantt et al. (2014) and Zhang et al. (2015) as described in detail in Zhang
et al. (2015). However, in WRF/Chem, the aerosol activation module is only linked to the
microphysics module through the variable CDNC, which is read by the microphysics module. It



is not coupled to the cumulus parameterization scheme unlike in WRF-CAM5 and CESM. The
FN05 series has been incorporated into module_mixactivate.F in the physics directory in
WRF/Chem. As BN07 involves the entrainment effect for convective clouds and has very small
impacts on non-convective CDNC (Zhang et al., 2015), it is not included in this study. In addition,
unlike Gantt et al. (2014) and Zhang et al. (2015), the KU09 treatment is also not included in this
study as the empirical constants $A_{FHH}$ and $B_{FHH}$ used in the formulation, which are compound-
specific, have not been experimentally determined for black carbon, although those constants have
been determined for dust and confirmed by Laaksonen et al. (2016). The additional MN14
treatment incorporated in this study involves a small modification to the original FN05 series
parameterizations (without KU09), and helps to better account for the size of inertially limited
CCN, and to remove a discontinuity in the calculation of the surface area of cloud droplets
(Morales Betancourt and Nenes, 2014). The updated treatments are about 20% more
computationally expensive to run as compared to ARG00 (Zhang et al., 2016), but capture the
sensitivity of CDNC to all aspects of the aerosol with comparable accuracy to numerical parcel
models, which was shown to be an underlying reason for biases from ARG (Morales Betancourt
et al., 2014).
**4.   Results and Discussions**
**4.1. Sensitivity Simulations with VBS Treatments Coupled with CB05**
As listed in Table 4, a number of sensitivity simulations are designed to identify the best model

configuration for OA treatments with the closest agreement to observations over CONUS. Those
sensitivity simulations consider (i) two SOA modules (MADE/SORGAM vs MADE/VBS), (ii)
two types of VBS treatment for POA (nonvolatile POA vs. semivolatile POA), (iii) two $\Delta H_{vap}$
treatments (default vs. the semi-empirical $\Delta H_{vap}$ equation by Epstein et al. (2010)), (iv) three



476 different percentages of functionalization and fragmentation (FF) (10%, 25%, and 50%), (v) three

477 sets of POA emissions (default vs. 1.5 or 3 times the original NEI POA emissions), (vi) three

478 different gas-phase mechanisms (CB05, CB6, and SAPRC07), and (vii) two different aerosol

479 activation schemes (ARG00 vs. combinations of different aerosol activation schemes of the FN05

480 series: FN05, FN05/BA10, and MN14) All simulations except for CB05-SORG-DH contain the

481 VBS treatments for OA. CB05-SORG-DH and CB05-VBS-DH treat POA emissions as

482 nonvolatile. In addition, the impact of two different cumulus parameterization schemes: Grell –

483 Freitas (Grell and Freitas, 2014) and the Multi-scale Kain Fritsch (MSKF) (Zheng et al., 2016)

484 scheme were also tested.

485  Table 5 summarizes the main statistics for all sensitivity simulations in terms of mean obs,

486 mean sim, Corr, NMB, and NME for hourly OC and TC concentrations against IMPROVE and

487 hourly TC concentrations against STN, respectively, over the whole CONUS domain. Figure 1

488 compares the domain mean hourly averaged observed OC or TC concentrations based on

489 IMPROVE and STN with simulated concentrations calculated based on the ratios of OA/OC 1.4

490 and 2.1 for each sensitivity simulation. The domain mean hourly averaged obs OC concentration

491 is 0.88 µg m$^{-3}$ for IMPROVE, and the domain mean hourly averaged obs TC concentration is 1.03

492 µg m$^{-3}$ for IMPROVE and 2.71 µg m$^{-3}$ for STN. As shown in Figure 1, the simulation

493 CB05_SORG_DH with the default SOA module SORG largely underpredicts OC and TC with the

494 largest NMBs and NMEs and the lowest Corr as compared to all other simulations with a SOA

495 module based on the VBS method. The remaining VBS simulations significantly reduce the biases

496 and errors in OC and TC from CB05_SORG_DH and also improve the correlation. Compared to

497 CB05_SORG_DH, CB05_VBS_DH with nonvolatile POA seems to perform relatively well in

498 terms of NMBs and Corr against IMPROVE OC, IMPROVE TC, and STN TC.



Adding the semivolatile POA treatment with 1.5 times the NEI POA emissions

(CB05_POA_DH) reduces simulated OC and TC concentrations as compared to CB05_VBS_DH,

due to the loss of mass from the semivolatile POA. As the POA mass is reduced, less surface area

is available for SOA precursors to condense onto, resulting in decreased OA (thus decreased OC

and TC) for CB05_POA_DH. Using the semi-empirical correlation of Epstein et al. (2010) for

$\Delta H_{vap}$ increases the OC and TC concentrations (CB05_POA vs. CB05_POA_DH). Compared to

the default $\Delta H_{vap}$ of 30 kJ mol$^{-1}$ used in CB05_POA_DH, the semi-empirical correlation of Epstein

et al. (2010) gives much higher $\Delta H_{vap}$ values, resulting in more of the organic vapors in the

particulate phase than in the gas phase. Compared to CB05-POA, the simulations with various FF

treatments decrease the OA concentrations, as part of the OA mass is fragmented to higher

volatility bins. The 10%FF case (CB05_10%FF) does not differ significantly from the no FF case

(CB05_POA). However, increasing the percentage of FF (from 10% to 25%, then to 50%)

decreases the OA concentrations. The FF treatments, however, even if they are more representative

of actual SOA atmospheric formation processes, reduce the Corr slightly (compared to the cases

CB05_POA and CB05-10%FF). By doubling the POA emissions (from 1.5 to 3.0 times the

original POA NEI emissions) for the 25% FF case (CB05_FF25%_EM3), the predicted OC and

TC concentrations are closer to the observations. When evaluated against IMPROVE OC,

IMPROVE TC, and STN TC, among for simulations using CB05, the simulations

CB05_VBS_DH, CB05_POA, CB05_FF10%, and CB05_FF25%_EM3 perform better than other

cases. The differences in the OC and TC predictions from the simulations with different gas-phase

mechanisms will be discussed later in Section 2.

Figure 2 shows the spatial distributions of simulated OC and TC concentrations overlaid with

observed OC from IMPROVE and TC from STN for the case CB05_25%FF_EM3 for the two



OA/OC ratios. The model performs much better for IMPROVE OC with an OA/OC ratio of 2.1
as compared to 1.4, especially over eastern U.S. where the use of an OA/OC ratio of 1.4 results in
large overpredictions. However over the central U.S. and parts of the western U.S., the use of an
OA/OC ratio of 1.4 shows slightly better predictions of IMPROVE OC compared to the use of
OA/OC ratio of 2.1 that gives underpredictions of OC. On the other hand, the model performs
better for STN OC with an OA/OC ratio of 1.4 as compared to 2.1. The use of an OA/OC ratio of
1.4 gives better agreement with STN TC over eastern U.S. where the use of an OA/OC ratio of 2.1
results in large underpredictions of TC. Evaluation of OC and TC against IMPROVE and STN,
respectively, therefore depends heavily on the OA/OC ratio, which is site-specific. Therefore in
more rural sites (IMPROVE), the OA/OC ratio is more likely to be high (~2.1) with more
oxygenated OA, while in more urban sites (STN), the OA/OC ratio is more likely to be lower
(~1.4) due to fresher emissions and less oxidized species.
Figure 3 shows the POA/OA ratios for six sensitivity simulations. As mentioned earlier, the
observed ratio of POA/total OA is approximately 15% to 40% during the summer period over
various locations in the CONUS. As SOA concentrations from field campaigns are sparse at
different locations and at different time periods, the POA/OA ratio is used to evaluate the model's
capability to reproduce POA and SOA concentrations. The simulation CB05_SORG_DH with
default SORGAM SOA module largely overpredicts the POA/OA ratio, due to significant
underpredictions    of    SOA.    The    simulations    CB05_VBS_DH,    CB05_50%FF,    and
CB05_25%FF_EM3 with various VBS treatments all have POA/OA ratios that fall within the
range of 0.15 to 0.4, with lower POA/OA ratios over more rural areas and higher POA/OA ratios
over urban areas. CB05_VBS_DH, however, might give too high POA concentrations over the
western portion of the domain as it does not consider POA to be semivolatile. Considering





semivolatile POA, however, without considering the fragmentation and functionalization
processes in the simulation CB05_POA results in too low POA/OA ratio (< 0.1 over most areas).
Similarly, the CB05_FF25% case also results in a large portion of CONUS with POA/OA ratios
of < 0.1, due to the loss of POA mass. CB05_FF50%, however, predicts reasonable POA/OA
ratios, even with fragmentation/functionalization due to balanced loss of both POA and SOA mass
through fragmentation to higher volatility bins. The simulation CB05_FF25%_EM3 also improves
from CB05_FF25% by increasing the POA mass contributing to higher POA/OA ratios.
Figure 4 shows the observed and simulated temporal variations of SOA concentrations at the
two CalNex sites: Bakersfield and Pasadena in CA from May to June 2010 for the simulations
CB05_SORM_DH,   CB05_VBS_DH,   CB05_25%FF_EM3,   CB6_25%FF_EM3,   and
SAPRC07_25%FF_EM3. There are large underpredictions of SOA by all runs on some days (e.g.
May 15 – 16, June 2 – 6, June 13 – 14) likely due to missing SOA precursor emissions. Table 6
shows the statistics of the simulations presented in Figure 4. The results using CB6 and SAPRC07
gas-phase mechanisms will be discussed in section 4.2. The observed SOA was derived based on
the tracer method of Kleindienst et al. (2012) which contains some uncertainties. For example, it
assumes mass fraction of the tracers in secondary organic carbon is the same in the field as that in
the laboratory, and the tracers are assumed to be inert and are unlikely to undergo oxidation in the
atmosphere, which might not be the case. In addition, the SOA data from the CalNex campaign
only consider contributions from a small number of precursors including biogenic precursors (i.e.,
isoprene, α-pinene, and β-caryophyllene), and the anthropogenic precursors (i.e., toluene,
polycyclic aromatic hydrocarbons (PAHs) and methyl butenol (MBO)).
As shown in Figure 4 and Table 6, the simulation CB05_SORG_DH with the default
SORGAM SOA module significantly underpredicts observed SOA concentrations at both sites.
The model configuration of CB05_VBS_DH has been used in a number of WRF/Chem
simulations published in literature (e.g., Yahya et al., 2015a; Campbell et al., 2015; Wang et al.,
2015a, b). At Bakersfield, the simulation CB05_VBS_DH overpredicts the SOA concentrations
for almost all the days. The simulation CB05_25%FF_EM3, however, underpredicts the SOA
concentrations at Bakersfield, especially in June. The CB05_25%FF_EM3 case also shows low
SOA concentrations throughout May and June, without much variability in SOA concentrations,
likely due to underestimations of original POA emissions at Bakersfield. As the S/IVOC emissions
for CB05_25%FF_EM3 are a factor of 3 of the original POA emissions from NEI, if the original
POA emissions from NEI are underestimated, the S/IVOC emissions will be low, resulting in low
SOA concentrations due to low concentrations of condensable material. At Pasadena, both
CB05_VBS_DH and CB05_25%FF_EM3 overpredict the obs SOA from May 15[th] to May 30[th],
but are unable to capture the high SOA concentrations from 2[nd] to 6[th] June. The CB05_VBS_DH
case seems to perform better than the CB05_25%FF_EM3 case when observed SOA
concentrations are high. The results from this study are consistent with those from Baker et al.
(2015), which showed that measured $PM_{2.5}$ OC at Bakersfield is largely underpredicted compared
to Pasadena. Baker et al. (2015), however, attributed to the underpredictions of OC at Bakersfield
and Pasadena mainly to primary OC predicted by the baseline model, as compared to the Aerosol
Mass Spectrometer measurements, suggesting that OC is mostly secondary in nature in Pasadena.
In addition, as mentioned earlier, the simulated SOA from WRF/Chem does not consider
contributions from all the SOA precursors identified by their trace compounds (e.g., the biogenic
precursor, b-caryophyllene, and the anthropogenic precursor MBO, are not included in
WRF/Chem), which can help to account for the discrepancies between the simulated and observed
SOA concentrations.




4.2. Sensitivity of OA predictions to different gas-phase mechanisms

Figure 1 shows that CB05_FF25%_EM3 produces the highest OC and TC concentrations at
the IMPROVE sites, followed by CB6_FF25%_EM3 and SAPRC07_FF25%_EM3, while
CB6_FF25%_EM3 produces the highest TC concentrations at the STN sites. However, the
differences in domain-mean simulated OC and TC between the simulations with the three different
gas-phase mechanisms are small, compared to the differences in simulated OC and TC due to
differences in VBS treatments (e.g., nonvolatile vs. semivolatile POA). Figure 4 also shows that
there are not much differences between simulated SOA concentrations with different gas-phase
mechanisms at Bakersfield, but larger differences are found at Pasadena. . For example,
SAPRC07_25%FF_EM3    produces    much    higher    SOA    concentrations    compared    to
CB05_25%FF_EM3 and CB6_25%FF_EM3 at Pasadena on several days (e.g., June 6-8). Figure
5 shows the time series of hydroxyl radical (OH) mixing ratios as well as diurnal plots of OH and
hydroperoxyl radical ($HO_2$) at Pasadena from the CalNex field campaign. The time series of $HO_2$
is not shown due to irregularity of the observational data. The model is able to reproduce the
diurnal variation of OH radicals but significantly overpredicts the daytime and peak OH mixing
ratios, especially for CB05 and CB6. All gas-phase mechanisms underpredict OH mixing ratios at
night. Among all simulations, SAPRC07 produces the closest simulated OH mixing ratios
compared to CB05 and CB6 gives the largest overpredictions. Similarly, the $HO_2$ mixing ratios
are generally overpredicted by all gas-phase mechanisms with SAPRC07 performing the best. The
overpredictions in OH and $HO_2$ mixing ratios do not help explain the underpredictions of SOA for
several days at Pasadena where underpredictions of VOCs may be the main cause, which is
consistent with the findings of Baker et al. (2015).





Figure 6 shows spatial distributions of average concentrations of oxidants including ozone

($O_3$), OH, $HO_2$, as well as the OA species including anthropogenic SOA (ASOA), biogenic SOA
(BSOA), TSOA, and POA. SAPRC07-25%FF-EM3 produces the highest domain average $O_3$
mixing ratios but the lowest domain average OH+$HO_2$ mixing ratios while CB6-25%FF-EM3
produces the highest domain average and maximum OH+$HO_2$ mixing ratios but the lowest domain
average $O_3$ mixing ratios. These findings are mostly consistent from literature. For example,
maximum $O_3$ and OH mixing ratios over the Los Angeles area are higher for CB6 compared to
CB05, which are consistent with the results from Yarwood et al. (2010). SAPRC07 also generally
produces higher $O_3$ mixing ratios compared to CB05. However, average $O_3$ mixing ratios from
CB6 are expected to be higher than CB05 (rather than lower as shown in Figure 6), according to
the study from Nopmongcol et al. (2012) which showed higher $O_3$ mixing ratios over Europe for
January and July using the Comprehensive Air Quality Model with Extensions (CAMx). CB6 is a
relatively new gas-phase mechanism, there are very few studies that evaluated its performance
over a longer period of time, e.g., for the whole summer, and over CONUS. In addition, there are
other uncertainties in this study. For example, the emissions for CB05 are used for CB6, the
additional explicit VOC species in CB6 such as benzene and acetylene are not considered, which
can also contribute to $O_3$ formation. In addition, most locations in the U.S. in 2010 are considered
to by $NO_x$-limited with localized VOC-limited regimes from May to September (Campbell et al.,
2015), which means that $O_3$ formation is more likely to depend on $NO_x$ rather than VOC
concentrations.

Table 7 shows the statistics for maximum 1-hr and 8-hr $O_3$ mixing ratios evaluated against

CASTNET and AQS. CASTNET sites are mainly rural sites, while AQS consists of urban,
suburban, and rural sites. As expected, SAPRC07 consistently produces the highest maximum 1-



hr and maximum 8-hr O$_3$ mixing ratios and overpredicts at AQS sites with an NMB of ~16%.
However, SAPRC07 performs the best at CASTNET sites, as both CB05 and CB6 significantly
underpredict maximum 1-hr and maximum 8-hr O$_3$ mixing ratios. At CASTNET sites, CB6
performs the poorest with the largest underpredictions for both maximum 1-hr and 8-hr O$_3$ mixing
ratios. However, CB6 predicts higher maximum 1-hr and 8-hr O$_3$ mixing ratios at AQS sites, while
CB05 predicts the lowest maximum 1-hr and 8-hr O$_3$ mixing ratios at AQS sites. It is likely that
CB6 predicts higher O$_3$ mixing ratios at more VOC-limited sites in urban areas, while CB05
predicts higher O$_3$ mixing ratios at more NO$_x$-limited areas, due to the improvement in VOC
speciation in CB6 compared to CB05. Overall, however, CB05 has the highest Corr and the lowest
NMEs for CASTNET maximum 1-hr and AQS maximum 1-hr and 8-hr O$_3$ mixing ratios. For
PM2.5 concentrations, CB6 produces the best performance against IMPROVE (highest Corr,
lowest NMB and NME) while CB05 produces the best performance against STN (highest Corr
and lowest NME). All 3 cases perform poorly for PM10 against AQS, with large underpredictions
due to the non-consideration of the coarse mode inorganic species in MADE-VBS treatments.
Anthropogenic SOA (ASOA) concentrations are lower for CB6 and SAPRC07 compared to
CB05. This is likely partially due to the emissions which are mapped from CB05 to CB6 and
SAPRC07. The CB05 emissions are not likely to account for all anthropogenic VOC emissions in
CB6 and SAPRC07, resulting in lower ASOA concentrations for CB6 and SAPRC07 compared
to CB05. Biogenic SOA (BSOA) concentrations, however, are the largest for CB6, followed by
SAPRC07 and CB05. BSOA concentrations are likely the highest for CB6 due to the highest
OH+HO$_2$ mixing ratios for CB6. The more extensive VOC representation and high O$_3$ mixing
ratios for SAPRC07 also likely contribute to the high BSOA concentrations for SAPRC07





compared to CB05. However, overall, the total SOA (TSOA) and POA concentrations for all three
gas-phase mechanisms do not vary much, resulting in similar OA concentrations.
Figures 7 and 8 show the time series of simulated vs, observed OC from IMPROVE and
simulated vs, observed TC from STN at several representative sites over CONUS for the different
gas-phase mechanisms. In general, at IMPROVE sites, CB05 gives the highest OC concentrations
compared to CB6 and SAPRC07 most of the time, resulting in overpredictions of OC
concentrations, while CB6 and SAPRC07 perform better against IMPROVE OC. The
overpredictions of CB05 are likely due to overpredictions in ASOA (as CB05 produces the highest
ASOA concentrations compared to CB6 and SAPRC07 as shown in Figure 6). As these sites are
located in rural locations, the dominant SOA is likely to be BSOA, or downwind ASOA from more
urban areas. With the exception of Death Valley NP, CA, the model performs relatively well in
predicting IMPROVE OC concentrations. Simulations with all three gas phase mechanisms
overpredict OC concentrations over several days in May in Brigantine NWR, Death Valley and
Swanqwarter, but is able to predict several of the peaks in June. All three gas-phase mechanisms,
however, largely underpredict OC concentrations over Death Valley from May 21$^{st}$ to June 30$^{th}$.
As the Death Valley NP is a desert, the OC at Death Valley NP is most likely due to downwind
OC transported from upwind locations, for which the model is not able to capture due to
meteorological biases such as biases in wind fields. The differences between simulation results
from the gas-phase mechanisms are smaller for STN TC compared to IMPROVE OC, probably
due to similar elemental carbon (EC) concentrations for all gas-phase mechanisms, which can form
a significant percentage of TC. In general, all simulations with the three gas-phase mechanisms
also show similar trends (peaks and troughs) for simulated TC, likely due to influences from
meteorological parameters such as wind and precipitation. Overall, all three simulations are also



able to predict the magnitude and trends of STN TC concentrations relatively well. Similarly,
CB05 tends to produce the highest TC concentrations, however, CB6 also does produce the highest
TC concentrations for several days, for example, for some days in May in Washington, DC and
Tampa, FL, as well as in June in Liberty, KS, likely due to influences of BSOA where CB6
produces the highest concentrations as shown in Figure 6.
**4.3. Impact of Different VBS treatments on CDNC**
Table 7 shows the statistics for model evaluation for simulated CDNC against MODIS-derived
CDNC from Bennartz (2007) for May to June 2010. All simulations underpredict CDNC due likely
to underpredictions in PM and CCN concentrations and uncertainties and/or assumptions in the
derived CDNC based on MODIS retrievals of cloud properties by Bennartz (2007) (Zhang et al.,
2015). For example, Bennartz (2007) derived the CDNC from cloud optical depths and cloud
effective radius assuming adiabatically-stratified clouds. Among all simulations with CB05,
CB05_SORG_DH produces the lowest CDNC due to underestimated OA concentrations.
Increasing the OA concentrations helps to reduce the negative biases for CDNC. There are small
differences, however, among simulated CDNC with different VBS treatments for CB05 in CDNC
predictions, with similar Corr ~ 0.29, NMBs of ~-29% to -27% and NMEs of ~ 47%. Figure 9
shows the spatial differences in predictions in warm clouds between the several simulations and
the simulation CB05_VBS_DH. CB05_SORGAM_DH gives the lower CDNC than
CB05_VBS_DH, indicating that the VBS treatment in CB05_VBS_DH helps to increase CDNC
significantly. While other simulations with semivolatile POA treatments further increase domain
average CDNC comparing to CB05_VBS_DH, the differences between CDNC predictions from
those simulations and CB05_VBS_DH are quite similar. In general, CDNC with the semivolatile



POA cases are higher over western U.S. but lower over eastern U.S. due to decreases in column
OA concentrations for the semivolatile POA cases compared to CB05_VBS_DH over eastern U.S.
The large differences in CDNC predictions, however, are found between simulations with the
different gas-phase mechanisms. SAPRC07_25%FF_EM3 has the largest negative bias (NMB of
-52%) compared to all other simulations with CB05 and the simulation with CB6. Figure 10
compares the spatial plots for CDNC predictions for simulations with different gas-phase
mechanisms, as well as the surface spatial plots for total OA and inorganic $PM_{2.5}$ concentrations.
The simulation with SAPRC07 shows significantly lower CDNC over northeastern U.S.
comparing to CDNC predictions from the other two simulations. While all three simulations show
similar total OA concentrations, large differences are found for their total inorganic $PM_{2.5}$
concentrations, with SAPRC07 producing the lowest domain mean and maximum total inorganic
$PM_{2.5}$ concentrations. Compared to CB05 and CB6, the lower inorganic $PM_{2.5}$ concentrations
simulated with SAPRC07 are likely due to the low $OH+HO_2$ mixing ratios for SAPRC07 as shown
in Figure 6, resulting in a lower PM number concentration and lower cloud condensation nuclei
(CCN), thus lower CDNC.
**4.4. Sensitivity Simulations for Aerosol Activation Parameterizations**
Among all OA sensitivity simulations, the simulation CB05-25%FF-EM3 gives an overall best
performance in terms of OC, TC, $O_3$, $PM_{2.5}$, and CDNC evaluation, it is thus selected to test various
aerosol activation parameterizations. As listed in Table 4, four sensitivity simulations are designed
to test the FN05 series aerosol activation parameterizations with improved treatments comparing
to the default ARG00 aerosol activation parameterization. These sensitivity simulations include
the default ARG00, the FN05, the combination of FN05 and BA10, and the MN14. These
simulations use the MSKF scheme instead of the Morrison microphysics schemes in the previous



729 SOA runs as the MSKF scheme has a better correlation with MODIS CDNC as compared to the

730 Morrison microphysics scheme. Table 8 summarizes the model evaluation results against MODIS-

731 derived CDNC from Bennartz (2007). The simulation ARG00 underpredicts CDNC with an NMB

732 of -35%. The FN05 series helps to reduce the underpredictions of CDNC significantly, because

733 they in general give higher activation fractions compared to the ARG00 parameterization under

734 most atmospheric conditions (Ghan et al., 2011). The addition of BA10 to the FN05 takes into

735 account the effects of condensation on giant CCN, which reduces the CDNC predictions and leads

736 to a negligible underprediction of CDNC (with an NMB of -0.8%) compared to a slight

737 overprediction by the FN05 with an NMB of 7.1%. MN14, which revises the original population

738 splitting method in FN05 and BA10, slightly increases the CDNC to an NMB of 4.2% comparing

739 to the FN05/BA10 simulation. The trends in the predictions of FN05, BA10, and MN14 are

740 consistent with the reported bias of ~+8%, -10% and -3%, respectively, by Morales Betancourt

741 and Nenes (2014) against the CDNC concentrations simulated from the cloud parcel model.

742 However, the Corr and NME are worse with the FN05 series and MN14. The NMEs are almost

743 doubled for the FN05 series and MN14, compared to that from the default ARG00. Figure 11

744 compares the spatial distributions of the simulated CDNC in warm clouds from ARG00, FN05

745 series, and MN14 and the MODIS-derived CDNC from Bennartz (2007). As shown in Figure 11,

746 the lower Corr and higher NMEs for the FN05 series as compared to ARG00 shown in Table 9, as

747 compared to ARG00, are due to the large overpredictions over northeastern U.S. but

748 underpredictions over other parts of the domain. The simulated CDNC from the default ARG00

749 case is similar to that from Bennartz (2007) over eastern U.S., the underpredictions are mainly

750 over western U.S. and over the ocean because of the known bias when large CCN are not present

751 (Morales Betancourt et al., 2014). The simulations with the FN05 series increase CDNC where



CCN is high, i.e., over the northeastern U.S., resulting in overpredictions in CDNC over
northeastern U.S., and does not help to improve CDNC predictions over other parts of the U.S. as
well as over the ocean.
Figure 12 compares the simulated CCN and AOD from the CB05_25%FF_EM3 + MN14 case
with those derived from the MODIS. The model largely underpredicts CCN, especially over the
western part of the domain, which explains the large underprediction of CDNC also over the
western part of the domain. Condensation of the available water vapor occurs over CCN which are
concentrated over northeastern U.S., resulting in overpredictions of CDNC over northeastern U.S.
The lack of CCN over the ocean and the western part of the domain is related to the
underpredictions of AOD over the same areas. This indicates biases in number (and probably mass)
concentrations of column PM concentrations, especially over the ocean and western U.S. $PM_{2.5}$
and $PM_{10}$ observational data are available over the surface and are both underpredicted, however,
there are no observational data for column concentrations of $PM_{2.5}$ and $PM_{10}$  for evaluation.
Improving the spatial distribution and magnitude of emissions for PM species and precursors for
the model layers at the surface and above the surface can help improve AOD and CCN predictions,
therefore CDNC predictions.

**5. Summary and Conclusions**

Current regional air quality models including WRF/Chem have large uncertainties in modeling
OA and aerosol-cloud feedback mechanisms such as the aerosol activation process. Comparing to
the traditional OA method, the VBS treatment helps to improve OA predictions by reducing the
underpredictions of OA. By including a semivolatile POA treatment, using a semi-empirical
formation of Epstein et al. (2010) for $\Delta H_{vap}$, including 25% fragmentation and functionalization
as well as including additional S/IVOC emissions, the VBS treatment in WRF/Chem simulates the



atmospheric OA formation processes more realistically and can perform relatively well in predictions of OC and TC against IMPROVE and STN. POA/OA ratios for the CB05_25%FF_EM3 and CB05_FF50% treatments are within the range of POA/OA ratios of ~0.15 to 0.40 from literature. Compared to the simulation with default SORGAM SOA module, the simulations with various new VBS treatments also give better agreement with observed SOA at Bakersfield and Pasadena during the CalNex field campaign from May to June 2010. However, biases exist in those simulations with the VBS treatments due to several possible reasons, including underestimated POA emissions, underpredicted VOC concentrations, as well as differences in the SOA precursors used in the model and those contributing to the observed SOA concentrations. The simulations with different gas-phase mechanisms (i.e., CB05, CB6, and SAPRC07) produce in general different ASOA and BSOA concentrations. SAPRC07 produces the highest $O_3$ mixing ratios, while CB6 produces the lowest $OH + HO_2$ mixing ratios. CB6 also performs the best when evaluated against IMPROVE PM2.5 while CB05 performs the best when evaluated against STN PM2.5 concentrations. All 3 cases perform poorly against AQS PM10 evaluation. Due to the significant differences between $O_3$, OH, and $HO_2$ mixing ratios for the three gas-phase mechanisms, inorganic PM concentrations vary widely, especially between the carbon bond mechanisms (CB05 and CB6) and SAPRC07, resulting in significantly different predictions of CDNC. The CDNC predictions do not vary much among simulations with CB05 and different VBS treatments, for example, for simulations with nonvolatile vs. semivolatile POA, and with and without fragmentation and functionalization treatments. The simulation with SAPRC07 produces the lowest CDNC compared to those with CB05 and CB6, due to the lowest inorganic PM number and mass concentrations resulted from the lowest OH and $HO_2$ mixing ratios among all





simulations. CB05 gives the best performances when evaluated against CASTNET and AQS ozone
mixing ratios, STN $PM_{2.5}$ concentrations and against MODIS CDNC.
With the default ARG00 treatment in the model, in general, all simulations with VBS
treatments underpredict the MODIS-derived CDNC by Bennartz (2007). By including the FN05
series (i.e., FN05, FN05/BA10, and MN14), the underpredictions for CDNC are greatly reduced.
However, the correlation coefficient and errors are worse with the FN05 series, with large
overpredictions over the northeastern U.S., where CCN is high. The model performs poorly for
AOD and CCN, likely due to inaccuracies in spatial distribution and magnitudes of PM and PM
precursor emissions in the model layers at the surface and above the surface. The CDNC
predictions can be improved by improving AOD and CCN undepredictions over western U.S. and
over the ocean.
**Code and Data Availability**
The WRF/Chem v3.7.1 code used in this paper will be available upon request. The inputs
including the meteorological files, meteorological initial and boundary conditions, chemical initial
and boundary conditions, model setup and configuration, and the namelist setup and instructions
on how to run the simulations for a 1-day test case, as well as a sample output for a 1-day test, can
be provided upon request.
**Acknowledgments**

This study is funded by the National Science Foundation EaSM program (AGS-1049200) at
NCSU. The emissions are taken from the 2008 NEI-derived emissions for 2010 provided by the
U.S. EPA, Environment Canada, and Mexican Secretariat of the Environment and Natural
Resources (Secretaría de Medio Ambiente y Recursos Naturales-SEMARNAT) and National
Institute of Ecology (Instituto Nacional de Ecología-INE) as part of the Air Quality Model





Evaluation International Initiative (AQMEII). The authors acknowledge use of the WRF-Chem
preprocessor tool mozbc provided by the Atmospheric Chemistry Observations and Modeling Lab
(ACOM) of NCAR and the script to generate initial and boundary conditions for WRF based on
CESM results provided by Ruby Leung, PNNL. This work also used the Stampede Extreme
Science and Engineering Discovery Environment (XSEDE) high-performance computing support
which is supported by the National Science Foundation grant number ACI-1053575. The authors
also acknowledge high-performance computing support from Yellowstone (ark:/85065/d7wd3xhc)
provided by NCAR's Computational and Information Systems Laboratory, sponsored by the
National Science Foundation and Information Systems Laboratory.

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





Tables

Table 1. Summary of several literature studies of VBS treatments in various regional and global models.

| Source | Ahmadov et al., 2012 | Shrivastava et al., 2011 | Farina et al., 2010 | Jathar et al., 2011 | Hodzic et al., 2010 |
|---|---|---|---|---|---|
| **Model** | WRF-Chem | WRF-Chem | GISS II' GCM | GISS II' GCM | CHIMERE |
| **Domain** | CONUS | Mexico City | Global | Global | Mexico City |
| **Spatial Resolution** | 20 km and 60 km | Nested 3 km within 12 km | $4^\circ \times 5^\circ$ | $4^\circ \times 5^\circ$ | 5 km × 5 km and 35 km × 35 km |
| **Emissions of SVOCs, IVOCs, and VOCs** | Only VOCs; no emissions of SVOCs and IVOCs | SVOC emissions 3 times POA emissions for both anthropogenic and biomass burning emissions. IVOC emissions 1.5 times POA emissions | POA is treated as nonvolatile and nonreactive, but acts as absorbing phase for SOA condensation, forming 1 OA phase | SVOC emissions are represented by the traditional emission inventory. IVOC emissions are 1.5 times traditional emissions | SVOC emissions 3 times POA emissions. IVOC emissions 1.5 times POA emissions |
| **No. of VBS bins** | 4 | 2 and 9 | 4 | 9 | 9 |
| **Aging** | Yes and No Simulations with aging: each oxidation step produces 7.5% additional mass | Yes and No Simulations with aging: each oxidation step produces 15% additional mass | Yes | Yes and No. Each oxidation step does not produce any additional mass | Yes. 2 cases below: (i)Each oxidation step produces 7.5% additional mass (ii)Each oxidation step reduces the volatility by 2 orders of magnitude and 40% of additional mass produced |
| **Observational data** | SEARCH, STN, IMPROVE | MILAGRO 2006 field campaign | IMPROVE, EMEP | IMPROVE, FAME, MILAGRO, SOAR | MILAGRO 2006 field campaign |
| **Variables evaluated** | OC and EC | OA, HOA, OOA O:C ratio | OM:OC of 1.8 | OA (surface), HOA, OOA | HOA, OOA, BBOA, O:C ratio |
| **Summary of results with VBS framework with/without aging compared to the traditional SOA approach** | -Improved diurnal variability -Results without the aging process underestimate OC throughout the day | -HOA and OOA: Lower negative bias with addition of S/IVOC emissions -OOA: 2 bin VBS better results than 9 bin VBS -Underprediction of O:C ratio in both cases | -IMPROVE: improved with aging -EMEP: aging further biases already high OA predictions | -Adding IVOC emissions improves performance, however underprediction remains in winter months | -HOA overpredicted during nighttime -Case(i):Modeled O:C 3 times lower than observed -Case(ii):Better agreement for O:C but SOA generally overestimated |

Note: HOA: Hydrocarbon-like OA – Reduced specie of OA, generally understood as a surrogate for urban combustion-related POA; OOA: Oxygenated OA – Characterized by its high oxygen content and generally understood as a surrogate for SOA; BBOA: Biomass burning OA





Table 1. (cont). SSummary of several literature studies of VBS treatments in various regional and global models.

| Source | Bergstrom et al., 2012 | Lane et al., 2008 | Donahue et al., 2009 | Murphy et al., 2009 | This work |
|---|---|---|---|---|---|
| **Model** | EMEP | PMCAMx | PMCAMx | PMCAMx | WRF/Chem |
| **Domain** | Europe, a large part of the North Atlantic and Arctic areas | Eastern U.S. | Eastern U.S. | Eastern U.S. | CONUS with parts of Canada and Mexico |
| **Spatial Resolution** | 50 km × 50 km. | 36 km × 36 km | NA | 36 km × 36 km | 36 km × 36 km |
| **Emissions of SVOCs, IVOCs and VOCs** | VOCs are present. S/IVOCs are 2.5 times the POA emissions | Only VOCs; SVOCs and IVOCs not added | Additional IVOCs added but details are not given | IVOC emissions are 0.2 to 0.8 times the nonvolatile POA emission rates | S/IVOCs are 1.5 to 3 times the nonvolatile POA NEI emissions |
| **No. of VBS bins** | 4 for SOA components and 9 for POA | 4 | 9 | 10 | 4 for SOA components and 9 for POA |
| **Aging** | Yes and No. Each oxidation step produces 7.5% additional mass | Yes. No additional mass produced for each oxidation step. | Yes. No additional mass produced for each oxidation step. | Yes. No additional mass produced for each oxidation step. | Yes. Each oxidation step produces 7.5% additional mass |
| **Observational data** | CARBOSOL, SORGA, Gote-2005 | STN, IMPROVE | NA | STN, IMPROVE | STN, IMPROVE, field data |
| **Variables evaluated** | TC, OC | OA | POA, OPOA, SOA | OA | TC, OC, POA/OA |
| **Summary of results with VBS framework with/without aging compared to the traditional SOA approach** | -Addition of aging reactions improve summertime results but has little or negative consequences in wintertime -Deteriorations of model results with increased aging at urban influenced sites in southern Europe | -Addition of aging reactions overpredicts the OA concentrations in rural IMPROVE stations but improves the model performance in urban areas | -Aging results in better model predictions | -Slight overprediction with IMPROVE -Underprediction with STN | -Large improvements in predictions |

Note: TC: total carbon; OC: Organic carbon; OPOA: oxidized POA





Table 2. Summary of main characteristics of CB05, CB6, and SAPRC07 gas-phase mechanisms.

| | CB05-Cl[1] | CB6 | SAPRC07[2] |
|---|---|---|---|
| **No. of species** | 70 | 114 | 118 |
| **No. of reactions** | 156 | 218 | 599 |
| **Lumping method** | Lumped structure based on carbon bonds | Lumped structure based on carbon bonds | Lumped species based on their reactivity towards hydroxyl (OH) |
| **Kinetic Data for rate constants** | Mostly from IUPAC (Atkinson et al., 2005). NASA/JPL (Sander et al., 2003) values were used in some cases where IUPAC data was not available. | New information from IUPAC (Atkinson et al., 2010) and NASA (Sander et al., 2006) | Mainly from IUPAC (2006) and NASA (Sander et al., 2006). |
| **Photolysis data** | Mainly from SAPRC99 chemical mechanism. IUPAC (Atkinson et al., 2005) was used if it differs significantly from SAPRC99. | New information from IUPAC (Atkinson et al., 2010) and NASA (Sander et al., 2006) | Mainly from IUPAC (2006) and NASA (Sander et al., 2006). |
| **Ozone chemistry** | Slightly underpredict $O_3$ mixing ratios with isoprene and in synthetic urban mixtures in chamber experiments. | Reduced underprediction in $O_3$ mixing ratios from benzene, toluene, and xylene, but forms $O_3$ from isoprene too slowly compared to CB05. | Slightly underpredict $O_3$ mixing ratios at low $NO_x$ levels in chamber experiments. |
| **Organic nitrate** | 2 reactions involving organic nitrate (NTR). | Additional $NO_x$ recycling from organic nitrate to represent fate of $NO_x$ over multiple days. | Added peroxy+NO reactions to form organic nitrate. |
| **Chlorine chemistry** | 20 additional reactions for Cl chemistry involving species $Cl_2$, HOCl, Cl, ClO, and FMCl. | CB05 chlorine chemistry included in this work. | 22 base chlorine reactions involving $CL_2$, CLNO, CLONO, $CLNO_2$, $CLONO_2$, HOCl, and 26 additional reactions involving organic products |
| **Organic chemistry** | - Explicit organic aerosol precursors, e.g., isoprene, toluene, xylene, α-pinene, ß-pinene. | - Explicit long-lived and abundant organic compounds including propane, acetone, benzene and acetylene added<br>- Extensive revision of isoprene and aromatics chemistry<br>-Formation of alpha-dicarbonyl compounds (glyoxal, glycoaldehyde, methylglyoxal)<br>- Updates to peroxy radical chemistry that will improve formation of $H_2O_2$ | - Reformulated reactions of peroxy radicals so that effects of changes in $NO_x$ conditions on organic product formation is more accurately represented<br>- Most comprehensive representation of VOCs compared to other gas-phase mechanisms |
| **3-D host models** | Implemented into WRF/Chem v3.6.1 by Wang et al. (2014). Also available in WRF/Chem v3.7.1 | Implemented in CAMx by ENVIRON (2013) | Implemented in CMAQ (Carter, 2010) |
| **Reference** | Yarwood et al. (2005) | Yarwood et al. (2010)<br>ENVIRON (2013) | Carter (2010) |

[1] CB05 gas-phase mechanism with reactive chlorine chemistry (Yarwood et al., 2005)
[2] SAPRC07 uncondensed and expanded version C, which includes reactions for peroxy radical operators (Carter, 2010).





Table 3. Factors to calculate S/IVOC emissions from POA emissions from Shrivastava et al. (2011), May et al. (2013a, c) and newly calculated factors for this study.

| Log Ci* at 298K | Normalized fraction for stationary emissions based on anthropogenic emissions from Shrivastava et al. (2011) | Fraction for gasoline emissions from May et al. (2013a) | Fraction for biomass burning emissions from May et al. (2013c) | New calculated fraction for all sources based on Shrivastava et al. (2011), May et al. (2013a, c), and % distribution of NEI emissions |
|---|---|---|---|---|
| -2 | 0.04 | 0.14 | 0.2 | 0.1754 |
| -1 | 0.02 | 0.13 | 0.0 | 0.0141 |
| 0 | 0.03 | 0.15 | 0.1 | 0.0961 |
| 1 | 0.05 | 0.26 | 0.1 | 0.1084 |
| 2 | 0.07 | 0.15 | 0.2 | 0.1799 |
| 3 | 0.11 | 0.03 | 0.1 | 0.0949 |
| 4 | 0.16 | 0.02 | 0.3 | 0.258 |
| 5 | 0.20 | 0.01 | 0.0 | 0.0249 |
| 6 | 0.32 | 0.11 | 0.0 | 0.0483 |





Table 4. Configurations for OA and aerosol activation sensitivity simulations. All simulations except for CB05-SORG-DH contain the VBS treatments for OA.

| Name | Gas-Phase | $\Delta H_{vap}$ | VBS | FF | POA emissions | Aerosol activation | Cumulus Scheme |
|---|---|---|---|---|---|---|---|
| CB05-SORG-DH | CB05 | 30 kJ mol$^{-1}$ | - | - | Original NEI | ARG00 | Grell-Freitas |
| CB05-VBS-DH | CB05 | 30 kJ mol$^{-1}$ | SOA | - | Original NEI | ARG00 | Grell-Freitas |
| CB05-POA-DH | CB05 | 30 kJ mol$^{-1}$ | SOA/POA | - | 1.5× | ARG00 | Grell-Freitas |
| CB05-POA | CB05 | Epstein et al. (2010) | SOA/POA | - | 1.5× | ARG00 | Grell-Freitas |
| CB05-50%FF | CB05 | Epstein et al. (2010) | SOA/POA | 50% | 1.5× | ARG00 | Grell-Freitas |
| CB05-10%FF | CB05 | Epstein et al. (2010) | SOA/POA | 10% | 1.5× | ARG00 | Grell-Freitas |
| CB05-25%FF | CB05 | Epstein et al. (2010) | SOA/POA | 25% | 1.5× | ARG00 | Grell-Freitas |
| CB05-25%FF-EM3 | CB05 | Epstein et al. (2010) | SOA/POA | 25% | 3.0× | ARG00 | Grell-Freitas |
| CB6-25%FF-EM3 | CB6 | Epstein et al. (2010) | SOA/POA | 25% | 3.0× | ARG00 | Grell-Freitas |
| SAPRC07-25%FF-EM3 | SAPRC07 | Epstein et al. (2010) | SOA/POA | 25% | 3.0× | ARG00 | Grell-Freitas |
| CB05-25%FF-EM3 (FN05) | CB05 | Epstein et al. (2010) | SOA/POA | 25% | 3.0× | FN05 | MSKF |
| CB05-25%FF-EM3 (FN05/ BA10) | CB05 | Epstein et al. (2010) | SOA/POA | 25% | 3.0× | FN05/ BA10 | MSKF |
| CB05-25%FF-EM3 (MN14) | CB05 | Epstein et al. (2010) | SOA/POA | 25% | 3.0× | MN14 | MSKF |

The suffix "DH" in the case names refer to cases with the default $\Delta H_{vap}$ of 30 kJ mol$^{-1}$, otherwise with the semi-empirical correlation by Epstein et al. (2010). The simulations without the suffix "POA" indicate cases with nonvolatile default POA emissions. The suffix "POA" in the case names refer to cases with semivolatile POA. The suffix "FF" in the case names refer to cases with semivolatile POA and with fragmentation and functionalization treatments, and the suffix "EM3" in the case names refer to cases with 3 times the original NEI POA emissions to take into account for missing S/IVOC species. "-" indicates not applicable.

Table 5. Range of statistics for OA/OC ratios of 1.4 and 2.1 (1.4/ 2.1) for May to June 2010. All
simulations use the ARG00 aerosol activation scheme and the Grell-Freitas cumulus parameterization.

| Case | Mean Obs | Mean Sim | Corr | NMB (%) | NME (%) |
|---|---|---|---|---|---|
| OC against IMPROVE | | | | | |
| CB05-SORG-DH | 0.88 | 0.28/ 0.19 | 0.26 | -68.1/ -78.7 | 73.9/ 80.9 |
| CB05-VBS-DH | 0.88 | 1.19/ 0.79 | 0.51 | 34.9/ -10.1 | 75.5/ 52.3 |
| CB05-POA-DH | 0.88 | 0.89/ 0.59 | 0.51 | 1.1/ -32.6 | 52.4/ 59.0 |
| CB05-POA | 0.88 | 1.05/ 0.70 | 0.51 | 18.9/ -20.7 | 63.2/ 49.2 |
| CB05-10%FF | 0.88 | 1.05/ 0.70 | 0.51 | 19.4/ -20.4 | 63.0/ 49.1 |
| CB05-25%FF | 0.88 | 0.86/ 0.57 | 0.49 | -2.9/ -35.2 | 54.6/ 51.4 |
| CB05-50%FF | 0.88 | 0.56/ 0.37 | 0.45 | -36.4/ -57.6 | 54.4/ 62.6 |
| CB05-25%FF-EM3 | 0.88 | 1.09/ 0.73 | 0.47 | 23.8/ -17.5 | 65.9/ 50.2 |
| CB6-25%FF-EM3 | 0.88 | 1.06/ 0.71 | 0.48 | 20.5/ -19.6 | 49.4/ 63.7 |
| SAPRC07-25%FF-EM3 | 0.88 | 1.00/ 0.67 | 0.46 | 13.3/ -24.4 | 60.1/ 50.4 |
| TC against IMPROVE | | | | | |
| CB05-SORG-DH | 1.03 | 0.44/ 0.34 | 0.30 | -57.6/ -66.7 | 67.9/ 72.3 |
| CB05-VBS-DH | 1.03 | 1.34/ 0.94 | 0.52 | 30.6/ -8.0 | 70.3/ 51.1 |
| CB05-POA-DH | 1.03 | 1.13/ 0.83 | 0.52 | 10.2/ -18.7 | 58.5/ 48.7 |
| CB05-POA | 1.03 | 1.29/ 0.94 | 0.53 | 25.6/ -8.5 | 63.8/ 48.3 |
| CB05-10%FF | 1.03 | 1.29/ 0.94 | 0.53 | 25.9/ -8.2 | 63.8/ 48.2 |
| CB05-25%FF | 1.03 | 1.09/ 0.83 | 0.51 | 6.8/ -21.6 | 55.2/ 48.2 |
| CB05-50%FF | 1.03 | 0.80/ 0.61 | 0.47 | -22.0/ -40.2 | 50.8/ 53.4 |
| CB05-25%FF-EM3 | 1.03 | 1.32/ 0.97 | 0.49 | 29.7/ -5.7 | 50.7/ 66.9 |
| CB6-25%FF-EM3 | 1.03 | 1.30/ 0.95 | 0.50 | 27.2/ -7.3 | 65.2/ 50.0 |
| SAPRC07-25%FF-EM3 | 1.03 | 1.23/ 0.90 | 0.48 | 20.6/ -11.9 | 61.4/ 49.4 |
| TC against STN | | | | | |
| CB05-SORG-DH | 2.71 | 1.34/ 1.10 | 0.29 | -50.6/ -59.4 | 60.1/ 64.9 |
| CB05-VBS-DH | 2.71 | 3.35/ 2.44 | 0.47 | 23.7/ -5.8 | 53.1/ 42.0 |
| CB05-POA-DH | 2.71 | 2.88/ 2.19 | 0.47 | 6.2/ -19.0 | 45.5/ 41.6 |
| CB05-POA | 2.71 | 3.03/ 2.30 | 0.46 | 11.7/ -15.3 | 44.6/ 39.9 |
| CB05-10%FF | 2.71 | 3.03/ 2.30 | 0.46 | 11.8/ -15.3 | 44.5/ 39.8 |
| CB05-25%-FF | 2.71 | 2.66/ 2.05 | 0.44 | -1.8/ -24.3 | 41.5/ 42.0 |
| CB05-50%-FF | 2.71 | 2.07/ 1.65 | 0.39 | -23.8/ -39.1 | 43.9/ 49.4 |
| CB05-25%FF-EM3 | 2.71 | 3.27/ 2.45 | 0.41 | 20.5/ -9.5 | 49.7/ 41.7 |
| CB6-25%FF-EM3 | 2.71 | 3.39/ 2.45 | 0.34 | 24.9/ -6.4 | 54.8/ 45.5 |
| SAPRC07-25%FF-EM3 | 2.71 | 3.00/ 2.28 | 0.41 | 10.7/ -16.1 | 45.2/ 42.0 |



Table 6. Statistics for evaluation at Bakersfield and Pasadena sites. A bar chart of daily average obs vs. sim values can be found in Figure 4.

| Case | Mean Obs | Mean Sim | Corr | NMB (%) | NME (%) |
|---|---|---|---|---|---|
| **Bakersfield** | | | | | |
| **CB05-SORG-DH** | 0.51 | 5.9e-04 | -0.15 | -100 | 100% |
| **CB05-VBS-DH** | 0.51 | 0.67 | 0.41 | 32.5 | 62.0 |
| **CB05-25%FF-EM3** | 0.51 | 0.24 | -0.01 | -52.0 | 61.0 |
| **CB6-25%FF-EM3** | 0.51 | 0.28 | -0.04 | -45.8 | 59.0 |
| **SAPRC07-25%FF-EM3** | 0.51 | 0.24 | -0.16 | -53.1 | 63.0 |
| **Pasadena** | | | | | |
| **CB05-SORG-DH** | 0.63 | 0.04 | -0.07 | -94.0 | 94.0 |
| **CB05-VBS-DH** | 0.63 | 0.54 | 0.09 | -14.5 | 64.3 |
| **CB05-25%FF-EM3** | 0.63 | 0.54 | -0.2 | -14.4 | 66.2 |
| **CB6-25%FF-EM3** | 0.63 | 0.62 | -0.2 | -2.1 | 70.0 |
| **SAPRC07-25%FF-EM3** | 0.63 | 0.62 | 0.03 | -1.4 | 70.5 |





Table 7. Statistics for max 1-h and max 8-h $O_3$ for simulations with different gas-phases against CASTNET and AQS for May to June 2010.

| Case | Mean Obs | Mean Sim | Corr | NMB (%) | NME (%) |
|---|---|---|---|---|---|
| **CASTNET Max 1-h $O_3$** | | | | | |
| CB05-25%FF-EM3 | 51.8 | 43.3 | 0.54 | -16.3 | 21.9 |
| CB6-25%FF-EM3 | 51.8 | 41.9 | 0.52 | -19.1 | 24.1 |
| SAPRC07-25%FF-EM3 | 51.8 | 48.3 | 0.51 | -6.7 | 21.1 |
| **CASTNET Max 8-h $O_3$** | | | | | |
| CB05-25%FF-EM3 | 47.4 | 43.0 | 0.54 | -9.3 | 18.9 |
| CB6-25%FF-EM3 | 47.4 | 41.8 | 0.53 | -11.8 | 20.6 |
| SAPRC07-25%FF-EM3 | 47.4 | 47.9 | 0.50 | 1.0 | 19.8 |
| **AQS Max 1-h $O_3$** | | | | | |
| CB05-25%FF-EM3 | 51.0 | 49.9 | 0.55 | -2.1 | 18.2 |
| CB6-25%FF-EM3 | 51.0 | 51.5 | 0.43 | 1.0 | 20.8 |
| SAPRC07-25%FF-EM3 | 51.0 | 59.3 | 0.44 | 16.4 | 26.1 |
| **AQS Max 8-h $O_3$** | | | | | |
| CB05-25%FF-EM3 | 46.2 | 46.0 | 0.54 | -0.4 | 18.6 |
| CB6-25%FF-EM3 | 46.2 | 47.4 | 0.47 | 2.6 | 20.3 |
| SAPRC07-25%FF-EM3 | 46.2 | 53.7 | 0.46 | 16.3 | 25.4 |
| **IMPROVE $PM_{2.5}$** | | | | | |
| CB05-25%FF-EM3 | 4.9 | 3.8 | 0.64 | -22.0 | 40.6 |
| CB6-25%FF-EM3 | 4.9 | 4.1 | 0.65 | -16.5 | 39.6 |
| SAPRC07-25%FF-EM3 | 4.9 | 3.5 | 0.60 | -28.5 | 42.9 |
| **STN $PM_{2.5}$** | | | | | |
| CB05-25%FF-EM3 | 11.1 | 8.8 | 0.48 | -20.6 | 40.7 |
| CB6-25%FF-EM3 | 11.1 | 10.0 | 0.37 | -9.3 | 44.3 |
| SAPRC07-25%FF-EM3 | 11.1 | 7.7 | 0.40 | -30.5 | 45.2 |
| **AQS $PM_{10}$** | | | | | |
| CB05-25%FF-EM3 | 24.6 | 7.3 | 0.08 | -70.2 | 73.5 |
| CB6-25%FF-EM3 | 24.6 | 8.0 | 0.09 | -67.7 | 71.8 |
| SAPRC07-25%FF-EM3 | 24.6 | 6.9 | 0.09 | -71.9 | 74.8 |





Table 8. Statistics for model evaluation for simulated CDNC against MODIS-derived CDNC from Bennartz (2007).

| Case | Mean Obs | Mean Sim | Corr | NMB (%) | NME (%) |
|---|---|---|---|---|---|
| CB05-SORG-DH | 162.1 | 96.0 | 0.28 | -40.8 | 50.4 |
| CB05-VBS-DH | 162.1 | 106.0 | 0.28 | -34.6 | 50.6 |
| CB05-POA-DH | 162.1 | 115.0 | 0.29 | -29.1 | 47.4 |
| CB05-POA | 162.1 | 117.3 | 0.29 | -27.7 | 47.3 |
| CB05-10%FF | 162.1 | 117.1 | 0.29 | -27.8 | 47.2 |
| CB05-25%-FF | 162.1 | 116.4 | 0.29 | -28.2 | 47.3 |
| CB05-50%-FF | 162.1 | 114.7 | 0.29 | -29.2 | 47.4 |
| CB05-25%FF-EM3 | 162.1 | 116.2 | 0.29 | -28.3 | 47.3 |
| CB6-25%FF-EM3 | 162.1 | 110.4 | 0.30 | -31.9 | 47.3 |
| SAPRC07-25%FF-EM3 | 162.1 | 77.3 | 0.26 | -52.3 | 55.8 |

Table 9. Statistics for simulated CDNC for CB05-25%FF-EM3 against MODIS-derived CDNC from Bennartz (2007) for May to June 2010.

| Case | Mean Obs | Mean Sim | Corr | NMB (%) | NME (%) |
|---|---|---|---|---|---|
| ARG00 | 162.1 | 104.8 | 0.31 | -35.4 | 49.9 |
| FN05 | 162.1 | 173.8 | 0.26 | 7.1 | 93.0 |
| FN05/BA10 | 162.1 | 160.8 | 0.27 | -0.8 | 87.9 |
| MN14 | 162.1 | 168.9 | 0.27 | 4.2 | 89.6 |





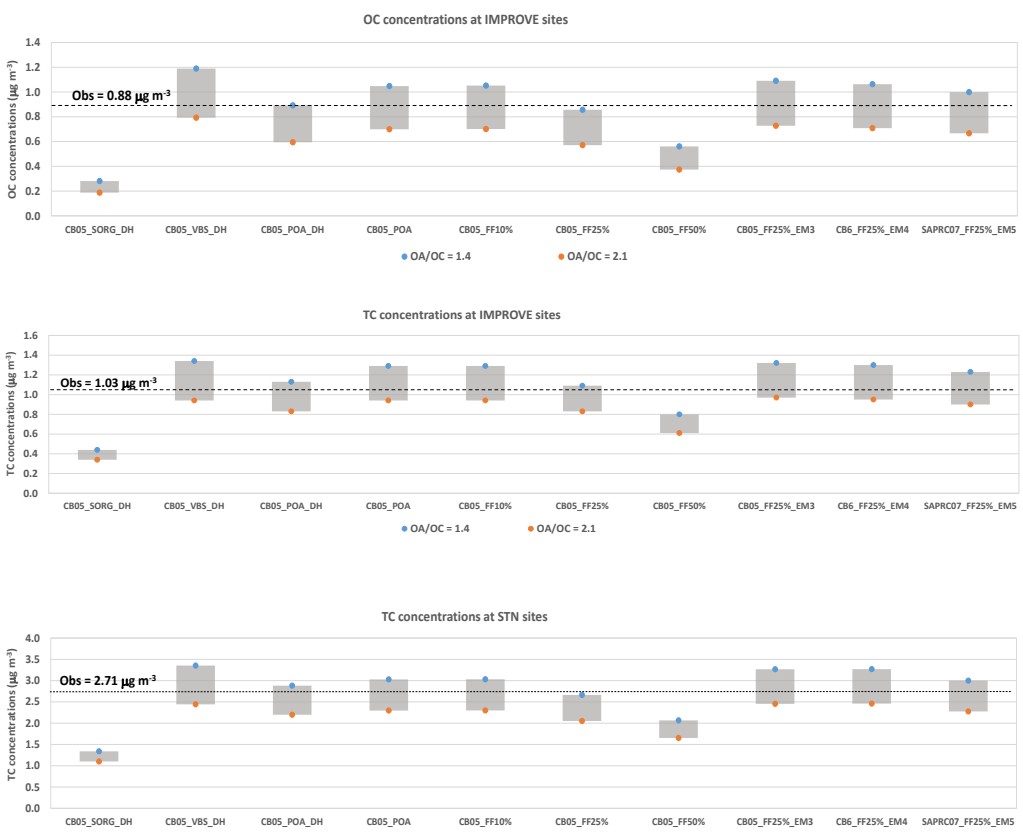

Figure 1. Sim OC and TC concentrations against observations from IMPROVE and STN under two A/OC ratios: 1.4 and 2.1, , resulting in a range of possible OC or TC values denoted by the gray bars. The obs OC or TC is denoted by the horizontal dotted line.





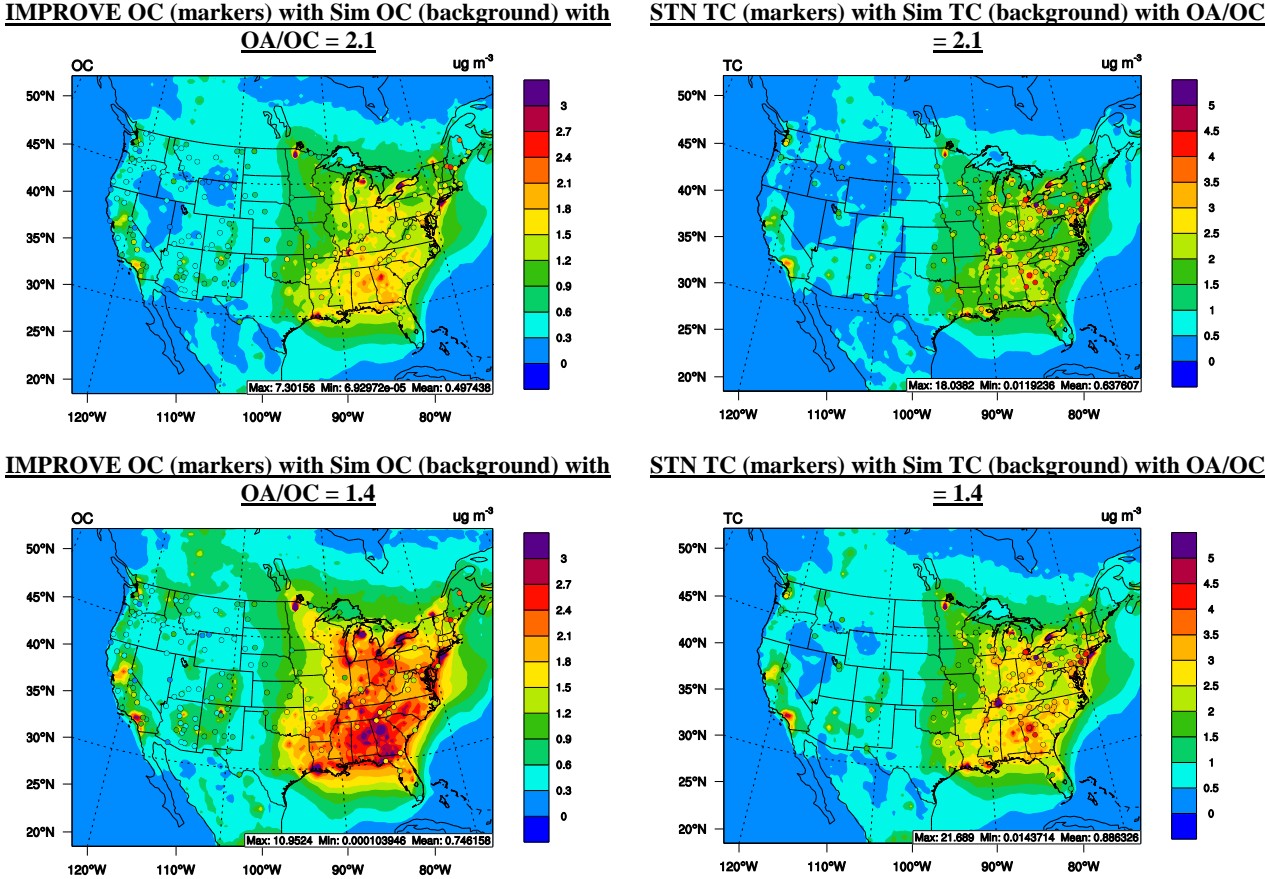

Figure 2. Overlay of obs data (markers) vs. sim data (background) for IMPROVE OC and STN TC and for OA/OC ratios of 1.4 and 2.1 for the case CB05_25%FF_EM3.





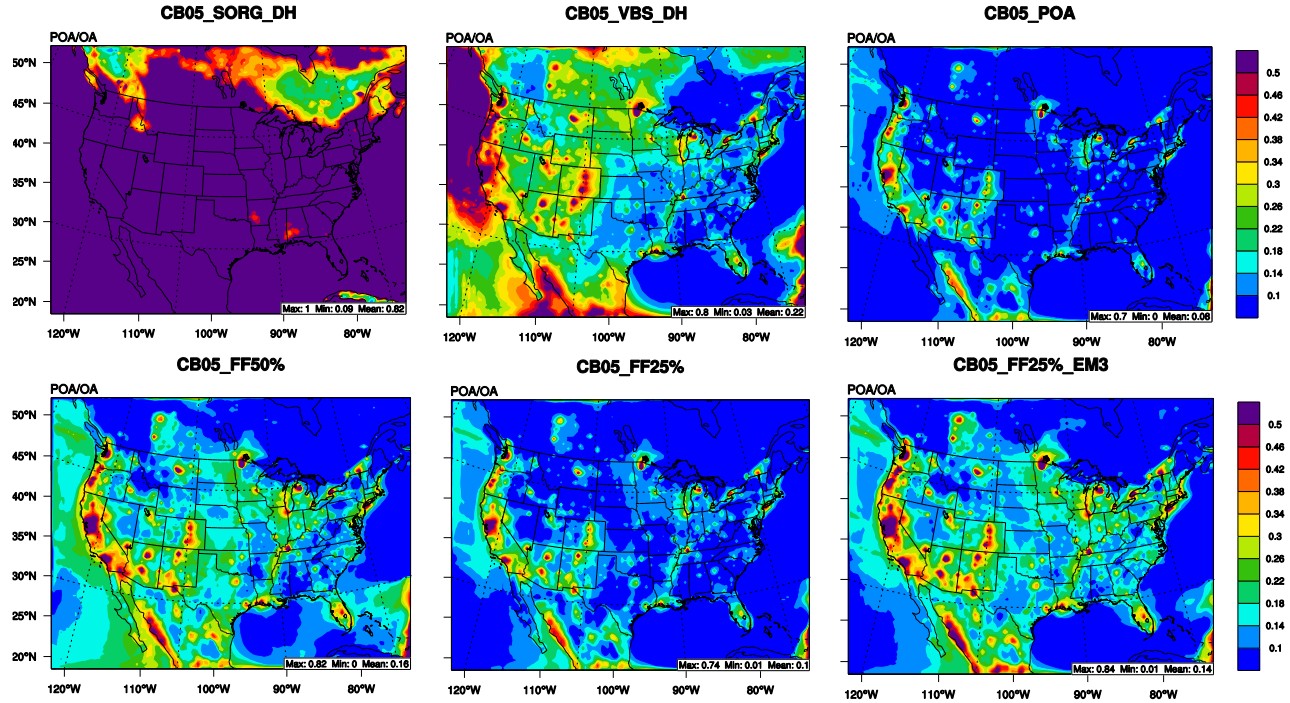

Figure 3. POA/OA ratios of sim data from various sensitivity simulation cases.





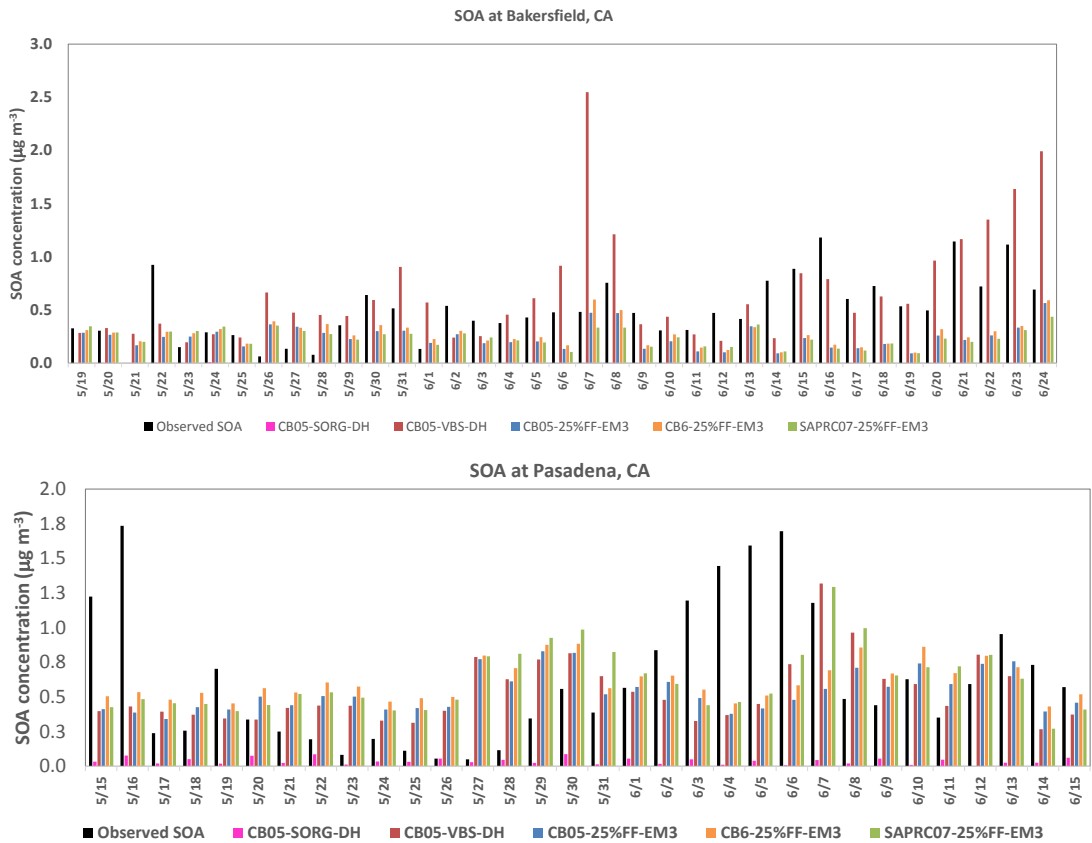

Figure 4. Comparison of obs SOA vs. sim SOA at CalNex sites in Bakersfield and Pasadena in
California.





Figure 5. Time series of OH and diurnal plots of OH and $HO_2$ at Pasadena, CA during CALNEX, 2010.





Figure 6. Spatial plots of several gas and aerosol species for the three cases with different gas-phase mechanisms.





Figure 7. Timeseries plots of IMPROVE OC vs. simulated OC at selected sites from sensitivity
simulations of different gas-phase mechanisms. The colored bands represent the range of OC values for
ratios 1.4 to 2.1.





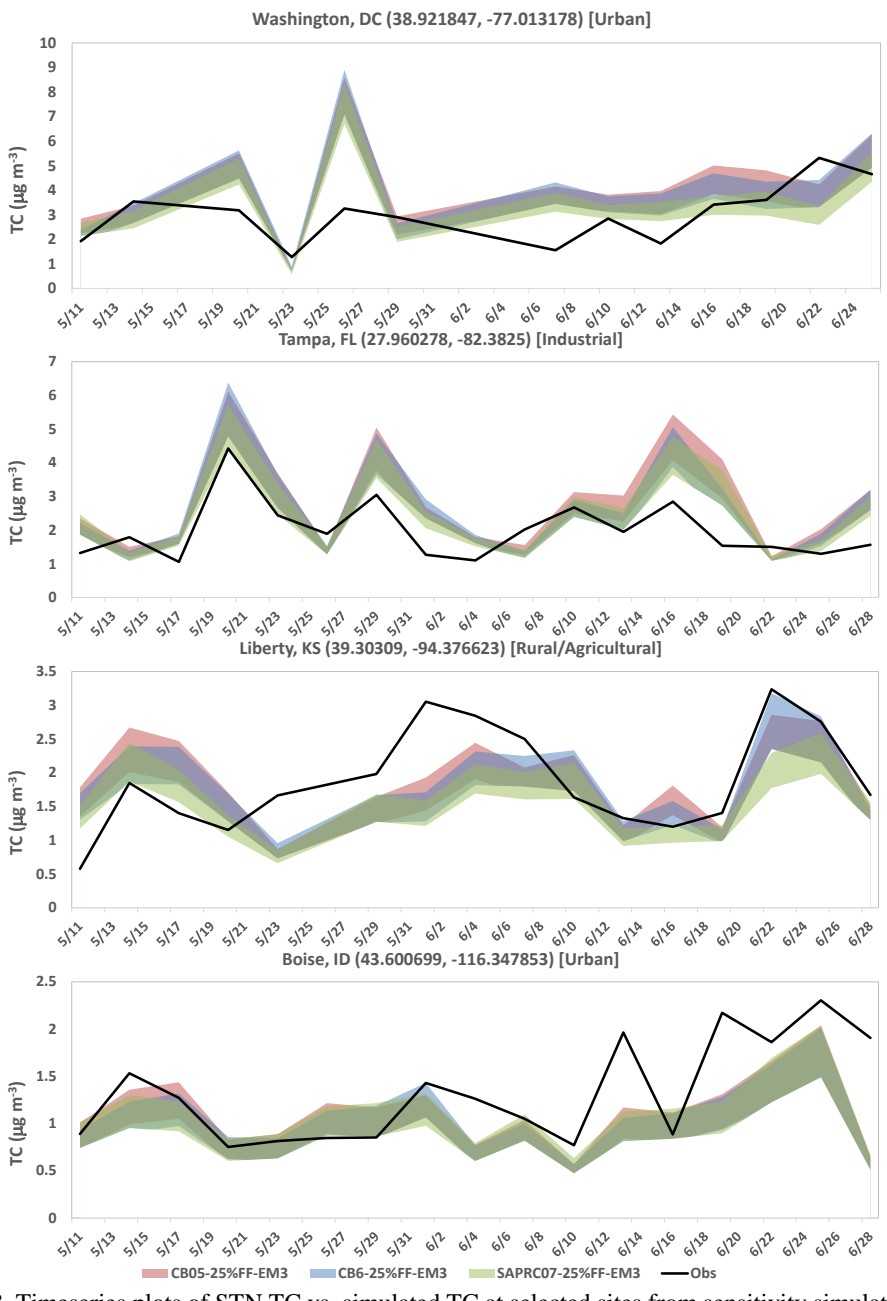

Figure 8. Timeseries plots of STN TC vs. simulated TC at selected sites from sensitivity simulations of different gas-phase mechanisms. The colored bands represent the range of OC values for ratios 1.4 to 2.1.





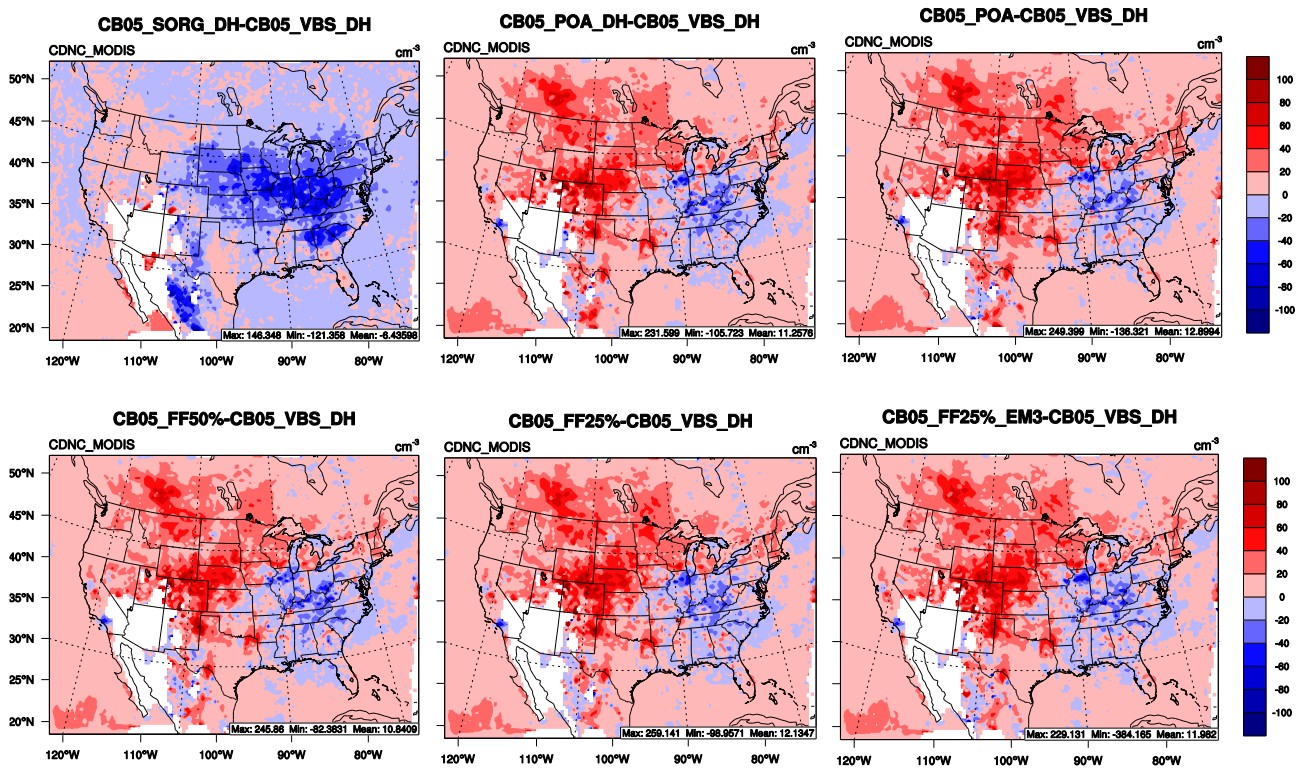

Figure 9. Impact of different VBS case on CDNC in warm clouds. The plots show the differences between the different sensitivity simulations and CB05_VBS-DH.





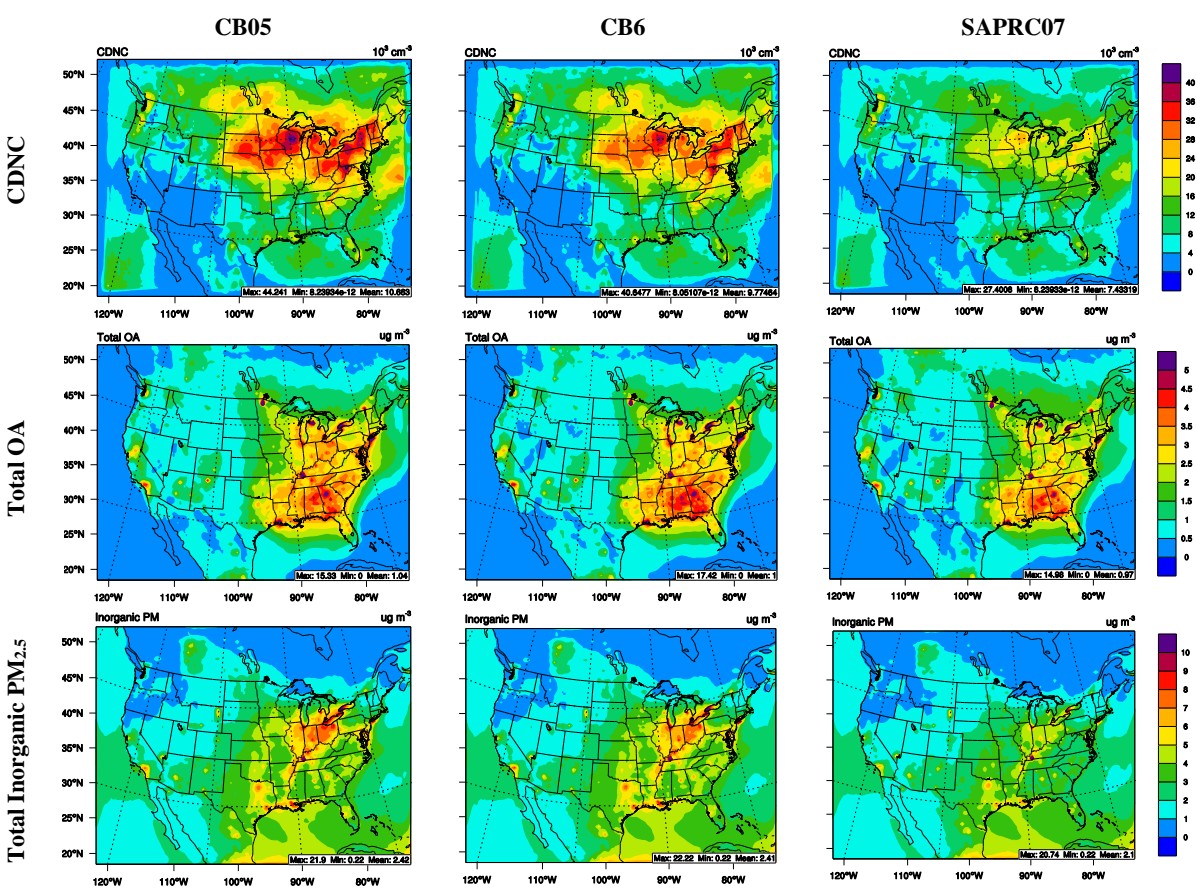

Figure 10. Spatial plots of CDNC, total surface OA and total inorganic PM$_{2.5}$ concentrations from different gas-phase mechanisms.





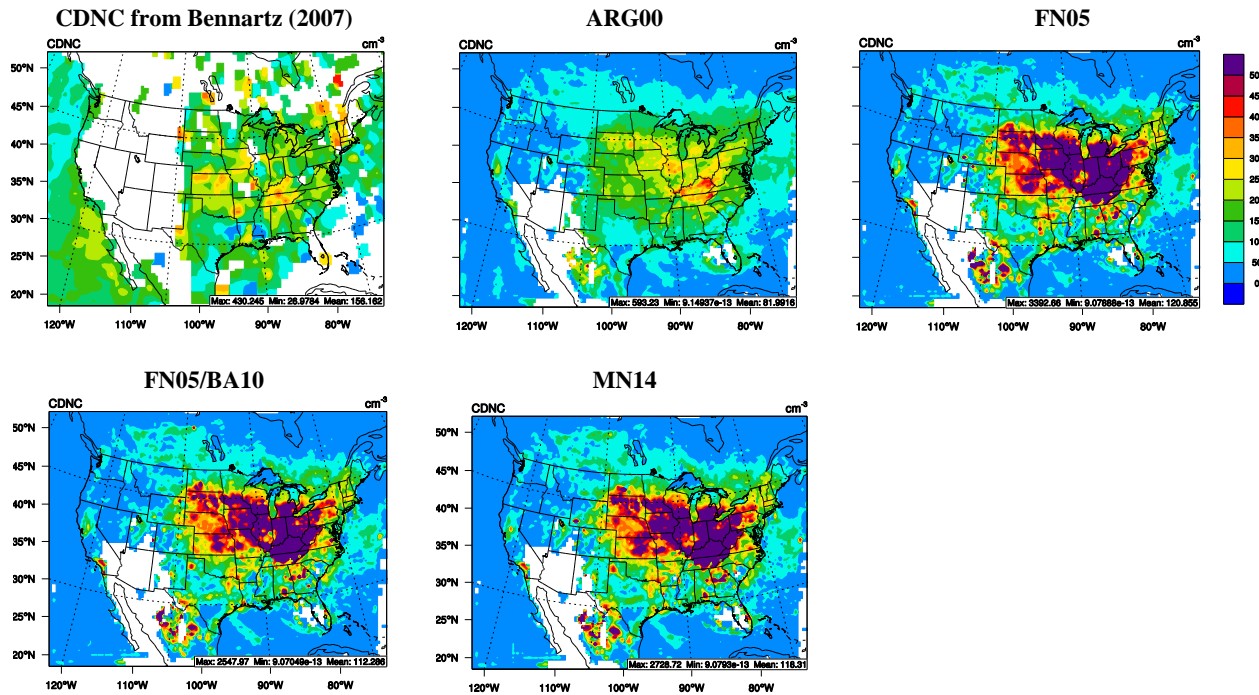

Figure 11. Spatial plots for MODIS-derived CDNC from Bennartz (2007) and simulated CDNC from CB05_25%FF_EM3 ARG00, FN series, and MN14 from May to June 2010.





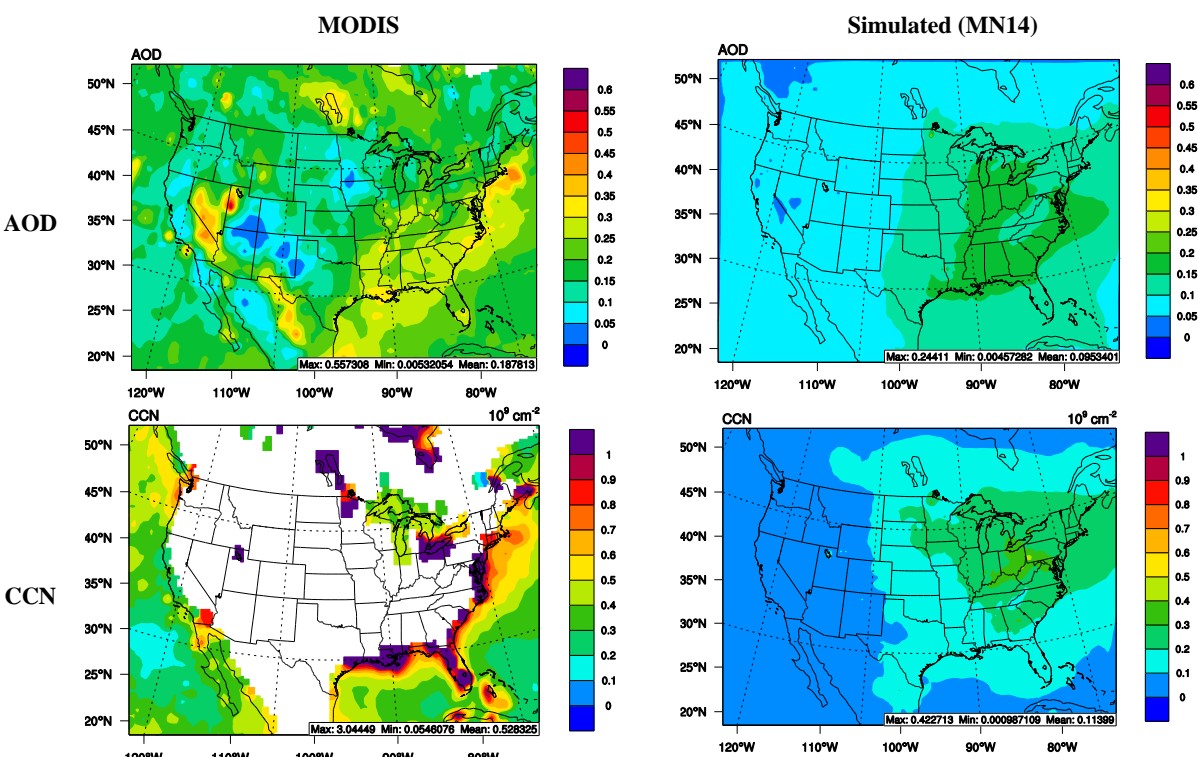

Figure 12. Plots of MODIS AOD and CCN against simulated AOD and CCN from MN14 with CB05_25%FF_EM3.