# Peer review of "Modeling Regional Air Quality and Climate: Improving Organic Aerosol and Aerosol Activation Processes in WRF/Chem version 3.7.1"

_Geoscientific Model Development, 2016_

## Referee Comment (RC1) · Anonymous Referee #1 · 21 Jan 2017

Regional models exhibit large uncertainties in the simulation of secondary organic aerosol (SOA) which have substantial impacts on climate due to aerosol-cloud inter-actions. This paper reviewed the current Volatility Basis Set (VBS) treatments and investigated the model performances in SOA simulation with a series of scenarios by changing the model configuration in chemical mechanisms and aerosol activation parameterization. Results suggest that simulations with VBS treatments present better agreement with observations compared to the traditional OA method, however, parameters such as the enthalpy of vaporization, percentage of fragmentation and functionalization, and POA emissions can largely influence the result. The paper is well written. I would recommend it to be published after minor revisions.

[Figure]

Apparently, the POA emissions play an important role in the simulation of SOA. Better performance is suggested in scenarios with increased POA emission. Does that imply that POA emission is underestimated in current NEI emissions? I would suggest the authors to provide some discussion about that.

Page 47: "SSummary" should be "Summary"

Page 50: Table 4. Note of "The simulations without the suffix "POA" indicate the cases with nonvolatile default POA emissions" need to be clarified, it should be "The simulations without the suffix "POA" or "FF"".

Page 52: Table 6. Poor correlation is suggested in most of cases, implying that some important SOA source is missing, biogenic SOA?

Page 53: for CASTNET, the simulated Max 8h O3 is very close to the simulated Max 1h O3, especially in CB6 (41.9 vs 41.8), but the observation doesn't (51.8 vs 47.4). Does that mean the model underestimate the peak value of O3?

Page 54: "CB05-25%FF-EM3" present different values in Table 8 and 9, while observation is the same. Please double check.

Page 55: Figure 1, "A/OC ratios" should be "OA/OC rations"

Page 64: Figure 10, it is very interesting that low CDNC shows at the edge of simulation domain, any explanation about that?

---

## Referee Comment (RC2) · Anonymous Referee #2 · 15 Mar 2017

The study is extensive, and is suitable for publication in GMD. I particularly like the summary of existing SOA approaches, centred around the VBS. This is a useful addition to the literature for sure. I request the following issues are addressed prior to publication.

General comments:

Page 4 line 80. I'm not sure the commonly held notion of computationally expensive SOA schemes according to the number of products should persist as a general discussion. Most, if not all, SOA models assume equilibrium absorptive mass partitioning which rests on Newton based methods requiring a small number of iterations. Is there a range of % contributions, for example, that display the relative cost of SOA schemes

versus, say, the gas phase chemistry?

The end of section 1.2 Would it be possible for the authors to comment on what conditions the activation schemes are initialised? Running at higher than 1km, presumably the assumption is to use the aerosol composition, both SOA and SIA, at a specific RH which is then fed into the ARG or FN schemes with regards to hygroscopicity? This might also impact the performance of any given activation scheme if the assumed mass is from a 'dry' SOA partitioning model, whereas SIA accounts for RH dependent partitioning.

Section 3.1: How does the new treatment of semi-volatile POA work with boundary conditions used to initialise simulations?

What is the impact of forcing different VBS profiles into one? I wasn't clear how this relates to, for example, the inputs required for the CCN schemes. Is it related to an inability to track separate sources through the simulation? Or is it related to how the emissions are convolved?

Section 4.3

Related to a previous point, the authors comment on how larger differences in CDNC predictions arise from different gas-phase mechanisms over VBS variants. I think it would benefit the reader, and the context of the sensitivity simulations to comment on how the VBS versus RH interactions feed into the CDNC parametrisations.

A more philosophical question, which doesn't require any modifications and isn't a critique of this study: I often wonder how much value we should place on assuming accurate ambient OA/OC measurements without going back to trailing the same model permutations in a controlled environment. Would the authors value smog chamber studies, on mixed VOC systems, using the same parametrisations but in a box-model configuration? It seems that, at least, this would be valuable from a high accuracy measurements perspective.

Minor comments:

Abstract line 41:'to 7.1%, it, however'. Please break the sentence here

page 14 line 316-317:'based on a number of literature', should be 'based on a number of studies in the literature'

page 29, lines 661-662:'simulated vs, observed' please replace this with 'simulated versus observed'

---

## Author Comment (AC1) · 16 May 2017

Reply to Reviewer 1

Regional models exhibit large uncertainties in the simulation of secondary organic aerosol (SOA) which have substantial impacts on climate due to aerosol-cloud interactions. This paper reviewed the current Volatility Basis Set (VBS) treatments and investigated the model performances in SOA simulation with a series of scenarios by changing the model configuration in chemical mechanisms and aerosol activation parameterization. Results suggest that simulations with VBS treatments present better agreement with observations compared to the traditional OA method, however, parameters such as the enthalpy of vaporization, percentage of fragmentation and functional-
ization, and POA emissions can largely influence the result. The paper is well written. I would recommend it to be published after minor revisions.

Reply: We thank the Reviewer for the comments to improve the presentation of the manuscript. Where applicable, suggestions have been taken into consideration and added to the manuscript. Please see below our point0-by-point replies.

Apparently, the POA emissions play an important role in the simulation of SOA. Better performance is suggested in scenarios with increased POA emission. Does that imply that POA emission is underestimated in current NEI emissions? I would suggest the authors to provide some discussion about that.

Reply: Yes, POA emissions are underestimated in current NEI emissions as POA is assumed to be nonvolatile. In the text, this sentence describes the underprediction in POA emissions: "With the semivolatile POA and FF cases in this study, additional IVOC and SVOC emissions are added as three times of the traditional POA emissions from NEI, to account for missing IVOC and SVOC species in the traditional POA emission inventory."

Page 47: "SSummary" should be "Summary"

Reply: This has been modified.

Page 50: Table 4. Note of "The simulations without the sufiňĄx "POA" indicate the cases with nonvolatile default POA emissions" need to be clarified, it should be "The simulations without the sufiňĄx "POA" or "FF"".

Reply: This has been modified.

Page 52: Table 6. Poor correlation is suggested in most of cases, implying that some important SOA source is missing, biogenic SOA?

Reply: Yes, as mentioned in the text: "The SOA data from the CalNex campaign only consider contributions from a small number of precursors including biogenic precursors

(i.e., isoprene, -pinene, and -caryophyllene), and the anthropogenic precursors (i.e., toluene, polycyclic aromatic hydrocarbons (PAHs) and methyl butenol (MBO))". This discussion has been made clearer to include that this reason also likely contributes to the poor correlation.

Page 53: for CASTNET, the simulated Max 8h O3 is very close to the simulated Max 1h O3, especially in CB6 (41.9 vs 41.8), but the observation doesn't (51.8 vs 47.4). Does that mean the model underestimate the peak value of O3?

Reply: Yes, this is likely to be true. In addition, NMBs and NMEs for Max 1h O3 are higher compared to Max 8h O3, which means that the model is not predicting well the transient peak O3 concentrations.

Page 54: "CB05-25%FF-EM3" present different values in Table 8 and 9, while observation is the same. Please double check.

Reply: Table 8 cases use the Grell-Freitas cumulus parameterization scheme, while Table 9 use the MSKF scheme. This has been made clear in the table headers.

Page 55: Figure 1, "A/OC ratios" should be "OA/OC rations"

Reply: This has been modified.

Page 64: Figure 10, it is very interesting that low CDNC shows at the edge of simulation domain, any explanation about that?

Reply: This is likely due to the fact that there are no boundary conditions for CDNC.

Please also note the supplement to this comment:
http://www.geosci-model-dev-discuss.net/gmd-2016-288/gmd-2016-288-AC1-supplement.pdf

―――――――――――――――――

**Supplement:**

[revised manuscript text omitted]

**IMPROVE OC (markers) with Sim OC (background) with OA/OC = 2.1**

**STN TC (markers) with Sim TC (background) with OA/OC = 2.1**

**IMPROVE OC (markers) with Sim OC (background) with OA/OC = 1.4**

**STN TC (markers) with Sim TC (background) with OA/OC = 1.4**

[Figure]

Figure 2. Overlay of obs data (markers) vs. versus sim data (background) for IMPROVE OC and STN TC and for OA/OC ratios of 1.4 and 2.1 for the case CB05_25%FF_EM3.

[Figure]

Figure 3. POA/OA ratios of sim data from various sensitivity simulation cases.

[Figure]

Figure 4. Comparison of obs SOA versus sim SOA at CalNex sites in Bakersfield and Pasadena in California.

[Figure]

Figure 5. Time series of OH and diurnal plots of OH and $HO_2$ at Pasadena, CA during CALNEX, 2010.

[Figure]

Figure 6. Spatial plots of several gas and aerosol species for the three cases with different gas-phase mechanisms.

[Figure]

Figure 7. Timeseries plots of IMPROVE OC  versus simulated OC at selected sites from sensitivity simulations of different gas-phase mechanisms. The colored bands represent the range of OC values for ratios 1.4 to 2.1.

[Figure]

Figure 8. Timeseries plots of STN TC  versus simulated TC at selected sites from sensitivity simulations of different gas-phase mechanisms. The colored bands represent the range of OC values for ratios 1.4 to 2.1.

[Figure]

Figure 9. Impact of different VBS cases on CDNC in warm clouds. The plots show the differences between the different sensitivity simulations and CB05_VBS-DH.

[Figure]

Figure 10. Spatial plots of total column CDNC, total surface OA and total inorganic PM$_{2.5}$ concentrations from simulations with different gas-phase mechanisms.

[Figure]

Figure 11. Spatial plots for MODIS-derived CDNC from Bennartz (2007) and simulated in-cloud column CDNC from CB05_25%FF_EM3 ARG00, FN series, and MN14 from May to June 2010.

[Figure]

Figure 12. Spatial plots of MODIS CCN and AOD against simulated CCN and AOD from MN14 with CB05_25%FF_EM3.

---

## Author Comment (AC2) · 16 May 2017

Reply to Reviewer 2

The study is extensive, and is suitable for publication in GMD. I particularly like the summary of existing SOA approaches, centred around the VBS. This is a useful addition to the literature for sure. I request the following issues are addressed prior to publication.

Reply: We thank the Reviewer for the comments to improve the presentation of the manuscript. Where applicable, suggestions have been taken into consideration and added to the manuscript. We hope that we were able to answer all the reviewer's

questions adequately. Please see below our point0-by-point replies.

Page 4 line 80. I'm not sure the commonly held notion of computationally expensive SOA schemes according to the number of products should persist as a general discussion. Most, if not all, SOA models assume equilibrium absorptive mass partitioning which rests on Newton based methods requiring a small number of iterations. Is there a range of % contributions, for example, that display the relative cost of SOA schemes versus, say, the gas phase chemistry?

Reply: The new SOA schemes, such as the VBS, is more computationally expensive in comparison to the "traditional" SOA schemes in 3-D models, such as the Odum 2-product model. From our experience, the SAPRC07 scheme is the most expensive compared to the other gas-phase chemistry schemes due to the number of chemical equations. Therefore, the coupled SAPRC07 with the VBS SOA model is computationally most expensive option. Unfortunately, we did not record the actual computational cost for each of the SOA schemes or the gas-phase chemistry schemes.

The end of section 1.2 Would it be possible for the authors to comment on what conditions the activation schemes are initialised? Running at higher than 1km, presumably the assumption is to use the aerosol composition, both SOA and SIA, at a specific RH which is then fed into the ARG or FN schemes with regards to hygroscopicity? This might also impact the performance of any given activation scheme if the assumed mass is from a 'dry' SOA partitioning model, whereas SIA accounts for RH dependent partitioning.

Reply: The chemical initial and boundary conditions (ICONs/BCONs) come from the modified CESM/CAM version 5.3 with updates by Gantt et al. (2014), He and Zhang (2014), and Glotfelty et al. (2016). Only the SIA concentrations are present as ICONs/BCONs. A 10-day spin-up is also used for the SOA concentrations to stabilize.

Section 3.1: How does the new treatment of semi-volatile POA work with boundary

conditions used to initialise simulations? What is the impact of forcing different VBS profiles into one? I wasn't clear how this relates to, for example, the inputs required for the CCN schemes. Is it related to an inability to track separate sources through the simulation? Or is it related to how the emissions are convolved?

Reply: As mentioned above, other than the differences in POA emissions as mentioned in the text, there are no other differences in ICONs/BCONs used in all the VBS cases. The model is unable to track the use of different VBS profiles in 1 simulation. Different simulations would have to be run, each changing 1 parameter to understand how the particular change affects SOA concentrations and CCN. The standard model inputs apply for both the VBS schemes, and the CCN schemes. No other special model inputs are required to run the CCN scheme, as the CCN scheme is dependent on the aerosol concentrations from the chemistry schemes, and vice versa.

Section 4.3 Related to a previous point, the authors comment on how larger differences in CDNC predictions arise from different gas-phase mechanisms over VBS variants. I think it would benefit the reader, and the context of the sensitivity simulations to comment on how the VBS versus RH interactions feed into the CDNC parametrisations.

Reply: The evaluation of the performance of RH by the model is interesting but beyond the scope of this study. However, from previous research, for example, by Yahya et al. (2015, 2016), the model performs relatively well for RH. We also do not expect RH to vary much between the different VBS cases. To ultimately understand how RH impacts the VBS performance, we would have to artificially vary the RH as inputs to the model. This might work better as a box-model study, rather than for a 3d model, where RH is predicted, and is temporally and spatially varying.

References: Yahya, K., J. He, and Y. Zhang (2015), Multiyear applications of WRF/Chem over continental U.S.: Model evaluation, variation trend, and impacts of boundary conditions, J. Geophys. Res. Atmos., 120, 12748–12777, doi:10.1002/2015JD023819.

[Figure]

Yahya, K., Wang, K., Campbell, P., Glotfelty, T., He, J., and Zhang, Y.: Decadal evalua-
tion of regional climate, air quality and their interactions over the continental US using
WRF/Chem version 3.6.1, Geosci. Model Dev., 9, 671 – 695, doi:10.5194/gmd-9-671-
2016, 2016

A more philosophical question, which doesn't require any modifi̧cations and isn't a
critique of this study: I often wonder how much value we should place on assuming
accurate ambient OA/OC measurements without going back to trailing the same model
permutations in a controlled environment. Would the authors value smog chamber
studies, on mixed VOC systems, using the same parametrisations but in a box-model
confi̧guration? It seems that, at least, this would be valuable from a high accuracy
measurements perspective.

Reply: Smog chamber studies, as well as box-model configurations are definitely valu-
able. In our opinion, they offer complementary information to 3-D model testing. As
a matter of fact, many smog chamber and box-model studies were indeed carried out
first, before the incorporation of the derived parameterisations into a 3-D model such
as the WRF/Chem model in this study. While box model studies are confined to a
controlled environment, using the 3-D model in our case represents real atmosphere
yet it introduces other uncertainties to OA concentrations from other atmospheric vari-
ables, feedbacks and processes. Both smog chamber studies/box-model studies and
3-D model studies have their own purposes and strengths, and should be used when
resources are permitted.

Minor comments: Abstract line 41:'to 7.1%, it, however'. Please break the sentence
here

Reply: This has been modified.

page 14 line 316-317:'based on a number of literature', should be 'based on a number
of studies in the literature'

Reply: This has been modified.

page 29, lines 661-662:'simulated vs, observed' please replace this with 'simulated versus observed'

Reply: This has been modified.

Please also note the supplement to this comment:
http://www.geosci-model-dev-discuss.net/gmd-2016-288/gmd-2016-288-AC2-supplement.pdf